# High-salt diet inhibits tumour growth in mice via regulating myeloid-derived suppressor cell differentiation

Wei He [1,2,3,5], Jinzhi Xu [1,5], Ruoyu Mu[1,2], Qiu Li [2], Da-lun Lv[4], Zhen Huang [1], Junfeng Zhang [1✉], Chunming Wang [2✉] & Lei Dong [1✉]

High-salt diets are associated with an elevated risk of autoimmune diseases, and immune dysregulation plays a key role in cancer development. However, the correlation between high-salt diets (HSD) and cancer development remains unclear. Here, we report that HSD increases the local concentration of sodium chloride in tumour tissue, inducing high osmotic stress that decreases both the production of cytokines required for myeloid-derived suppressor cells (MDSCs) expansion and MDSCs accumulation in the blood, spleen, and tumour. Consequently, the two major types of MDSCs change their phenotypes: monocytic-MDSCs differentiate into antitumour macrophages, and granulocytic-MDSCs adopt pro-inflammatory functions, thereby reactivating the antitumour actions of T cells. In addition, the expression of p38 mitogen-activated protein kinase-dependent nuclear factor of activated T cells 5 is enhanced in HSD-induced M-MDSC differentiation. Collectively, our study indicates that high-salt intake inhibits tumour growth in mice by activating antitumour immune surveillance through modulating the activities of MDSCs.

---

[1] State Key Laboratory of Pharmaceutical Biotechnology, School of Life Sciences and Medical School of Nanjing University, 163 Xianlin Avenue, Nanjing 210093, China. [2] State Key Laboratory of Quality Research in Chinese Medicine, Institute of Chinese Medical Sciences, University of Macau, Taipa, Macau SAR. [3] Department of Immunology, School of Basic Medical Sciences, Anhui Medical University, Hefei 230032, China. [4] Department of Burn and Plastic Surgery, First Affiliated Hospital of Wannan Medical College, Jinghu District, Wuhu City, Anhui Province 241000, China. [5] These authors contributed equally: Wei He, Jinzhi Xu. ✉email: jfzhang@nju.edu.cn; cmwang@umac.mo; leidong@nju.edu.cn

Excessive intake of dietary salt (NaCl) is clearly defined as an unhealthy lifestyle, due to its strong correlation with higher risks of chronic inflammation[1], cardiovascular disease[2] and autoimmune diseases[3]. Therefore, many professional societies recommend an upper salt intake limit of 3.75–6 g per day[4]. Growing evidence suggests that high-salt intake causes dysfunction in innate and adaptive immune responses. For instance, a high-salt diet (HSD) has been shown to activate interleukin (IL)-17-producing helper T (Th17) cells in vivo and exacerbate autoimmune encephalomyelitis (EAE) and cerebrovascular diseases[3,5,6]. Meanwhile, a high level of NaCl weakens the suppressive capacity of regulatory T cells on immune reactions, and promotes the secretion of many inflammatory cytokines, such as interferon (IFN)-γ[7]. In addition, under high-salt stimulation, macrophages polarise towards a typical M1 phenotype, secreting more pro-inflammatory cytokines, reactive oxygen species (ROS) and nitric oxide synthase 2 (NOS2), and activating inflammasome-related pathways[8–11]. These findings suggest that high-salt intake promotes inflammation in various tissue microenvironments, which may induce inflammatory diseases such as rheumatoid arthritis (RA) and colitis[12,13]. However, in the tumour microenvironment, stimulating the pro-inflammatory activities of multiple cell types is key to overcoming the established immunosuppression and consequently restoring immune attack against the tumour.

Myeloid-derived suppressor cells (MDSCs) play essential roles in tumour-induced immune tolerance[14,15]. MDSC expansion in lymphoid organs (the spleen, bone marrow and so on), blood, and tumours has been detected in tumour-bearing mice and cancer patients[16]. MDSCs are immature myeloid cells (IMCs) that are classified into two major subsets: monocytic MDSCs (M-MDSCs) and granulocytic MDSCs (PMN-MDSCs)[17,18]. Both have strong immunosuppressive activities but distinct functional and biochemical characteristics[19]. Meanwhile, as myeloid-derived monocyte precursors, MDSCs are highly plastic[20]. In response to various circumstance factors, M-MDSCs can differentiate into macrophages or dendritic cells (DCs)[21], and both M-MDSCs and PMN-MDSCs can dramatically change from being immunosuppressive to immunostimulatory. The stimulation can be external factors, such as cytokines and extracellular membrane components, or internal signals, such as tumour growth factor (TGF)-β, sirtuin 1 and paired Ig-like receptor B[22–24]. As a typical environmental stress sensed by cells, hypertonicity induced by high-salt intake can trigger immune responses in macrophages mediated by transcription factors, such as the tonicity-responsive enhancer-binding protein (NFAT5)[25]. Studies have demonstrated that the p38/NFAT5 axis enhances the expression of a number of pro-inflammatory genes and the survival of macrophages[26,27]. However, whether high-salt intake similarly influences the functions of MDSCs in the body is unknown.

Based on the above evidence, we hypothesise that HSD can inhibit tumour progression through priming MDSCs in the tumour microenvironment into an immunostimulatory phenotype. To explore this possibility, we establish two allograft tumour models in mice given a HSD and examine the effect of the HSD on tumour growth. Interestingly, our results suggest that HSD, though recognised as an unhealthy dietary pattern, dramatically regulates the differentiation of MDSCs in the tumour niche, leading to the removal of local immune suppression and inhibition of tumour progression.

## Results

**HSD retarded allograft tumour growth in murine models.** To investigate the effect of HSD on tumour growth, we established two grafted tumour models in female wild-type C57BL/6 and BABL/c mice: a mouse melanoma model established by implantation of B16F10 cells and a mouse mammary cancer model established by implantation of 4T1 cells. After 1 day of starvation, mice were subcutaneously injected with 4T1 or B16F10 cells. These mice were then fed a normal-salt diet (NSD) or HSD. The HSD-treated mice of both tumour models showed a lower tumour weight (Fig. 1a) and size (Fig. 1b) than the corresponding NSD-treated mice. The overall survival of the mice is shown in Fig. 1c. HSD prolonged the survival of the tumour-bearing mice, and more than 50% of mice on HSD survived for more than 5 weeks, whereas NSD mice died within 6 weeks. Figure 1d shows the tumours harvested after 16 days of the different treatments. Histologic analysis further revealed that HSD caused large necrotic areas in the tumour tissues (Supplementary Fig. 1a, b). HSD was also effective in decreasing tumour size (Fig. 1a, b) and prolonging tumour-bearing survival (Fig. 1c) in male tumour-bearing mice, indicating that the effects of HSD were not limited by gender. In addition, we also examined the food and water intake of tumour-bearing mice during the HSD period, and found that HSD markedly increased the food and water intake of mice (Supplementary Fig. 2a, b). To our surprise, the body weights of tumour-bearing mice were not affected by HSD, but were consistently lower than the weights of tumour-free mice (Supplementary Fig. 2c). Finally, we also investigated whether HSD impacted the general health of the animals and found no marked differences in spleen, liver or kidney indexes between the NSD and HSD groups (Supplementary Fig. 3a–f). Meanwhile, mice fed the HSD for 16 days did not exhibit hepatotoxicity or nephrotoxicity, demonstrated by a lack of differences in the serum levels of alanine transaminase (ALT), aspartate aminotransferase (AST), blood urea nitrogen (BUN) or creatinine (Supplementary Fig. 3g–j). Together, the data suggested that HSD inhibited the growth of transplanted tumours in vivo.

**HSD-enhanced osmotic stress via increasing NaCl storage.** To analyse the mechanism by which HSD retarded tumour growth, we first evaluated the effects of HSD on $Na^+$, $K^+$ and $Cl^-$ storage and water content within different organs of the tumour-bearing mice by atomic absorption spectroscopy or titration. Although the $Na^+$ and $Cl^-$ contents were higher in the thymus, livers, spleens and tumours in HSD mice than in NSD mice of both grafted tumour models, with $Na^+$ storage at its highest in the tumours (Fig. 2a and Supplementary Fig. 4a), no differences were observed in the serum, hearts or lungs of mice fed with NSD or HSD for 16 days (Fig. 2a, b, Supplementary Fig. 4a, b). Furthermore, there were no marked differences in $K^+$ or water content within these organs and serum between the NSD and HSD groups (Supplementary Fig. 4c–e). We also examined $Na^+$, $K^+$ and $Cl^-$ storage in the tumours by X-ray fluorescence spectrometry, and found that HSD enhanced $Na^+$ and $Cl^-$ storage but did not affect $K^+$ content in the tumour tissue (Fig. 2c, Supplementary Fig. 4f). Sodium is essential for the distribution of body fluids and maintenance of osmotic stress. Therefore, we further investigated the effects of HSD on the osmotic pressure in these organs of tumour-bearing mice by vapour-pressure osmometry (Vapro Wescor 5520). In both tumour models, there were no significant differences in the osmolality of serum, hearts or lungs in mice fed with NSD or HSD for 16 days (Fig. 2d, Supplementary Fig. 4g). However, the thymus, livers, spleens and tumours were clearly hyperosmolar in mice with HSD. Remarkably, among these tissues, tumour tissues from mice with HSD had the highest hyperosmolality, consistent with the high NaCl accumulation within the tumour (Fig. 2d, Supplementary Fig. 4g), indicating that salt enhanced osmotic stress by accumulating sodium storage within tumour tissues. Together, these data

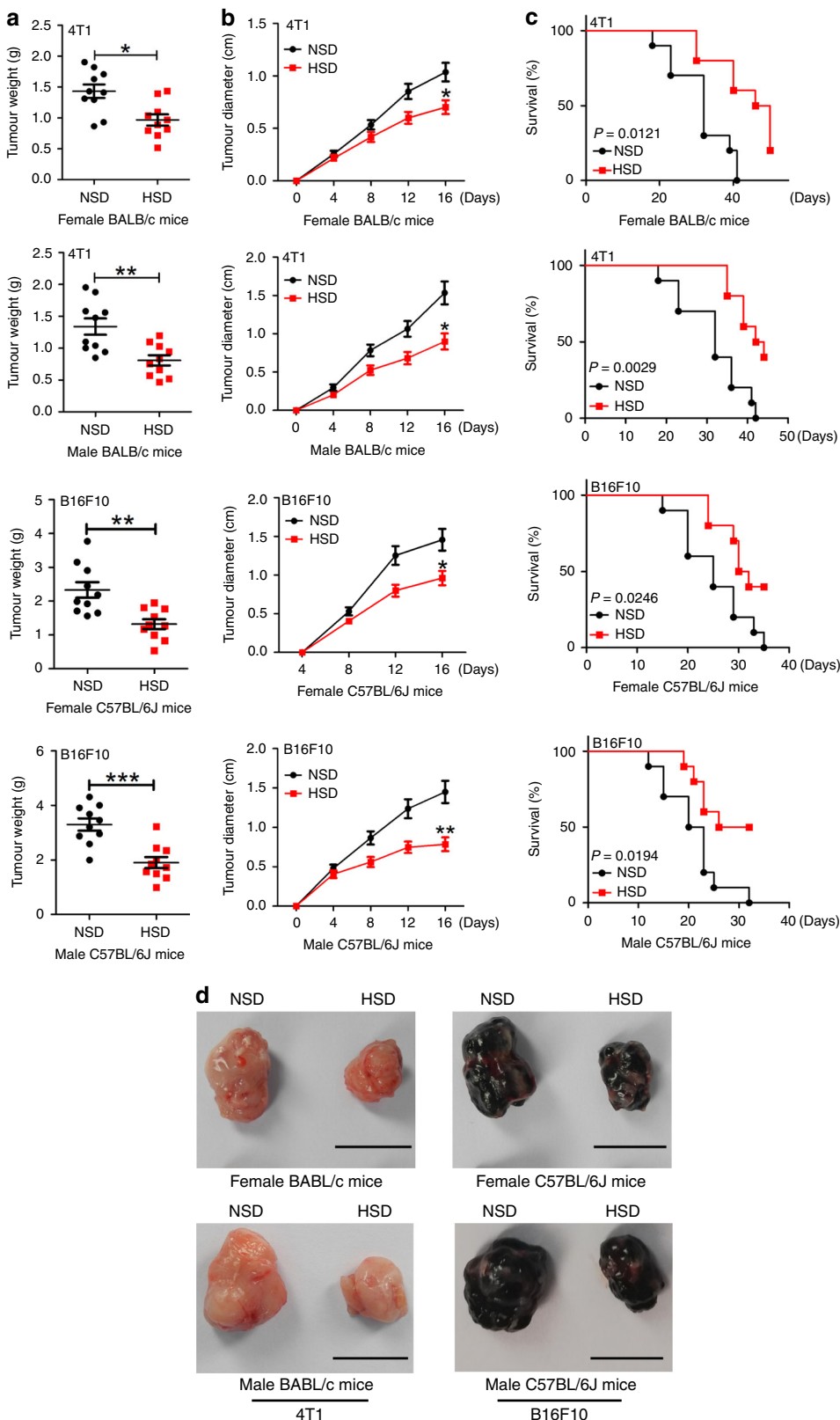

**Fig. 1 HSD retarded allograft tumour growth. NSD- or HSD-fed mice were subcutaneously injected with tumour cells until the tumour tissues were harvested. a** Tumour weight on day 16 post implantation. **b** Tumour growth curve from mice on the NSD and HSD for 16 days after tumour cells were engrafted. **c** Survival rate of tumour-bearing mice in each treatment cohort for more than 5 weeks post implantation. **d** Representative images of tumours harvested from NSD- and HSD-fed mice. Scale bar, 1 cm. For all panels, the two-tailed Wilcoxon rank-sum tests; $n = 10$ mice per group; *$p < 0.05$, **$p < 0.01$ and *** $p < 0.001$ vs. the NSD group. These experiments were replicated with similar results. Data are representative of three independent experiments, and are presented as the mean ± SEM. Source data are provided as a Source Data file.

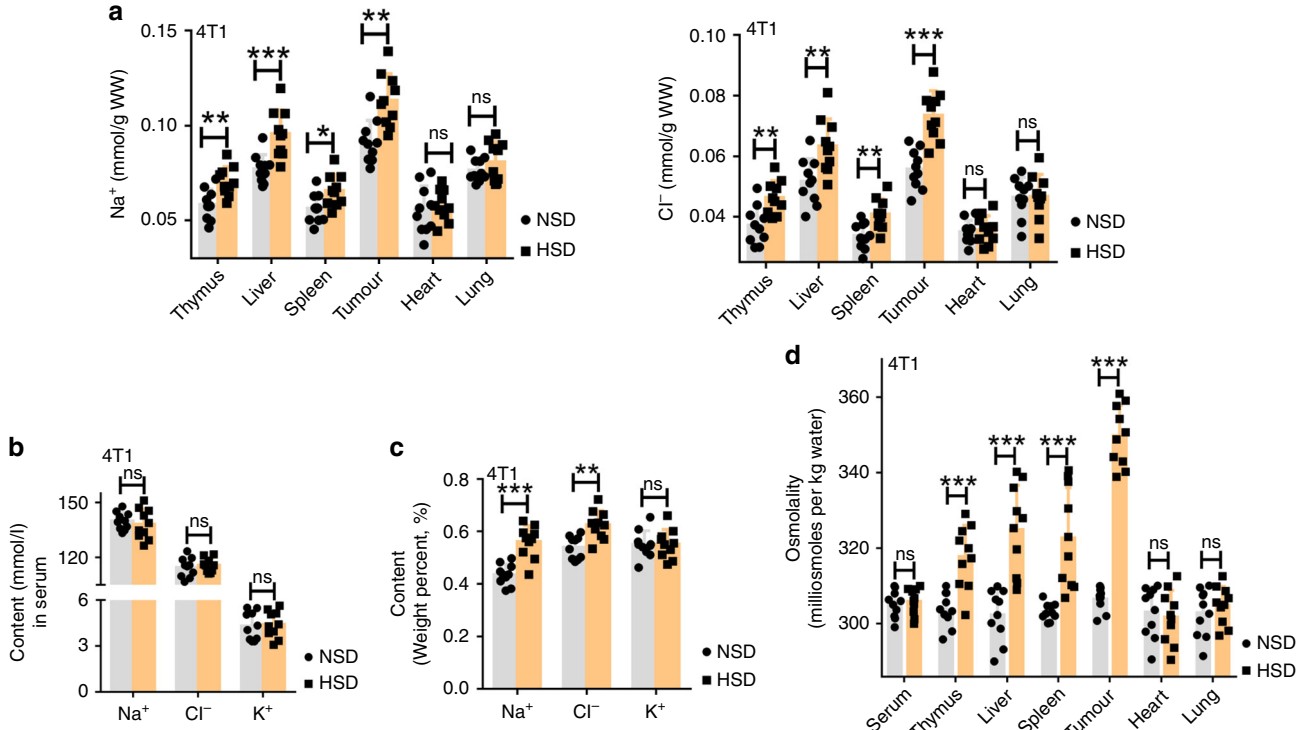

**Fig. 2 HSD increased tissue osmotic stress via promoting NaCl accumulation. a, b** Na$^+$ and Cl$^-$ content in different organs compared with plasma concentrations in the same mice when 4T1 tumour-bearing mice were fed the NSD and HSD for 16 days. **c** Na$^+$, K$^+$ and Cl$^-$ content in tumour tissues was determined by X-ray fluorescence spectrometry. **d** The osmolality of tissues from 4T1 tumour-bearing mice was examined by vapour-pressure osmometry. For all panels, the two-tailed Wilcoxon rank-sum tests; $n = 10$ mice per group; ns not significant; *$p < 0.05$, **$p < 0.01$ and ***$p < 0.001$. These experiments were replicated with a comparable number of mice three times. Data are representative of three independent experiments, and are presented as the mean ± SEM. Source data are provided as a Source Data file.

demonstrated that HSD resulted in the accumulation of the salt in some mouse tissues, especially tumour tissues, causing a significant hyperosmolality in tumours. The osmotic stress likely triggers subsequent cellular reactions in tumour tissue.

**HSD-induced pro-inflammatory reactions in vivo.** Emerging evidence suggests that high-salt conditions can modulate the functions and differentiation of immune cells[7,8,28]. To determine whether HSD influences the levels of the cytokines in serum and tumours, we measured cytokine levels in different tissues using a Proteome Profiler Mouse XL Cytokine Array (Ray Biotech). Compared with NSD, HSD significantly increased the production of several pro-inflammatory cytokines, including IL-12p40 and intracellular adhesion molecule (ICAM)-1, and decreased the level of IL-6, IL-10 and granulocyte–macrophage colony-stimulating factor (GM-CSF) in serum and tumour tissues from 4T1 tumour-bearing mice (Fig. 3a, Supplementary Fig. 5a). In addition, further verification of the level of IFN-γ and tumour necrosis factor (TNF)-α in tumour tissues by ELISA or qRT-PCR revealed that HSD also increased the level of IFN-γ and TNF-α in 4T1 tumours, as shown in Fig. 3b. Moreover, the results in the B16F10 tumour model (Supplementary Fig. 6a, b) were similar to those in the 4T1 model. These results suggested that the tumour growth-inhibiting effects of HSD might be due to changes in immune-system reactions.

**HSD promoted MDSC differentiation in vivo.** GM-CSF and IL-6 are known to inhibit myeloid precursor differentiation, inducing an increase in the number of MDSCs[29,30]. Therefore, we investigated whether the inhibitory effects of HSD on tumour

growth were due to changes in MDSC differentiation or function. We first studied the influence of HSD on the amount of MDSCs in the blood, spleen and tumour tissues from animals 16 days after B16F10 or 4T1 tumour implantation. As demonstrated by the data (Fig. 3c, Supplementary Fig. 7a–c), within CD45$^+$ cells, HSD markedly decreased the number of MDSCs (CD11b$^+$Gr-1$^+$ cells) in the blood, spleen and, particularly, tumours. We further analysed the change in the distribution of M-MDSCs (CD11b$^+$Ly-6C$^+$ cells) and PMN-MDSCs (CD11b$^+$Ly-6G$^+$ cells), and found significantly fewer M-MDSCs in these tissues in HSD-fed mice than in NSD-fed mice of both tumour models (Fig. 3d, Supplementary Fig. 7a–c), while little difference was observed in the percentage of PMN-MDSCs (Fig. 3e, Supplementary Fig. 7a–c). The change in the number of M-MDSCs suggested a possible differentiation from MDSCs towards macrophages or DCs; therefore, we investigated the presence of macrophages and DCs in the spleens and tumours. HSD significantly increased the number of macrophages (CD11b$^+$F4/80$^+$ cells) in the tumour tissue, but did not alter the number of macrophages in the spleen (Fig. 3f, g, Supplementary Fig. 7b, c). Moreover, there was no significant difference in the proportion of DCs (CD11b$^+$CD11c$^+$ cells) within CD45$^+$ cells from the NSD and HSD groups in the spleens or tumour tissues (Fig. 3f, g, Supplementary Fig. 7b, c). Recent studies have suggested that HSD may inhibit the potentially pro-tumour renin–angiotensin system (RAS)[31,32]; therefore, we performed additional experiments to investigate this possibility. HSD had significant antitumour effects under RAS inhibition using an angiotensin-converting enzyme inhibitor (captopril) (Supplementary Fig. 8a, b, d, e). Meanwhile, although both HSD and captopril decreased the level of plasma angiotensin II (AngII), the level of

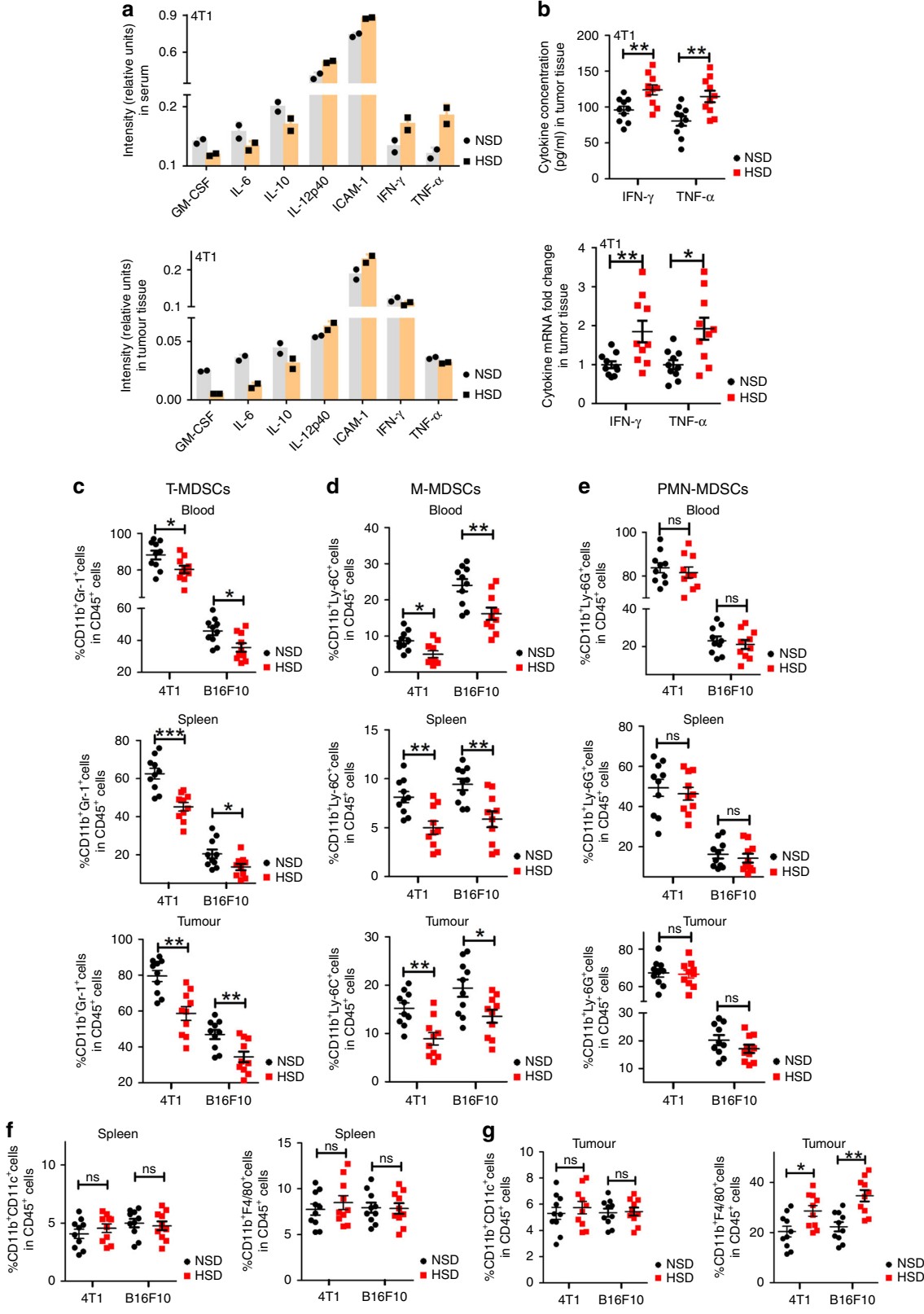

plasma AngII was lower in the captopril group than in the HSD group (Supplementary Fig. 8c, f), suggesting that the anti-tumour activity of HSD may be independent of RAS inhibition. Together, our results indicated that HSD might exert its anti-tumour effects by promoting the differentiation of M-MDSCs into macrophages.

**HSD-regulated MDSC differentiation and function**. The above data suggested that HSD-induced hyperosmolality might have an impact on M-MDSC differentiation and function. To address this, we isolated M-MDSCs from the tumour tissues harvested from 4T1 or B16F10 tumour-bearing mice NSD- or HSD-fed mice and analysed their differentiation. As shown in Fig. 4a, b, after being

**Fig. 3 The effect of HSD on inflammatory cytokine expression and MDSC differentiation in vivo. a** Tumour homogenates and serum samples were harvested and mixed at equal quantity or equal volumes. Inflammatory cytokines in tumour tissue lysates and serum from tumour-bearing NSD- or HSD-fed mice were assessed by the Proteome Profiler Mouse XL Cytokine Array, the signal intensity of the arrays was analysed using densitometry and the relative intensity (NSD vs. HSD) of individual cytokines was analysed after normalising to the positive controls on the same membrane. Each dot represents the technical replication of one phenotype. Data are from one out of two independent experiments. **b** ELISA and qRT-PCR of TNF-α and IFN-γ in tumour tissues from NSD- and HSD-fed mice. The two-tailed Wilcoxon rank-sum tests; $n = 10$ mice per group; *$p < 0.05$ and **$p < 0.01$ vs. the NSD group. Data are from one out of three independent experiments. Data are presented as dot plots extending to minimum and maximum values in one independent experiment, and bars are presented as the mean ± SEM of 10 individual mice. **c–g** 4T1 tumour-bearing mice were fed the NSD and HSD for 16 days, and the proportions of cells in blood, spleen and tumours were evaluated by flow cytometry. **c–e** The proportions of MDSCs in blood, spleen and tumours were evaluated by flow cytometry. **f, g** The proportion of DCs and macrophages in spleen and tumour from tumour-bearing mice are shown. The two-tailed Wilcoxon rank-sum tests; $n = 10$ mice per group; ns, not significant; *$p < 0.05$ and **$p < 0.01$ vs. the NSD group. Data are from one out of three independent experiments. Data are presented as dot plots extending to minimum and maximum values in one independent experiment, and bars are presented as the mean ± SEM of ten individual mice. Source data are provided as a Source Data file.

cultured for 2, 3 and 4 days, M-MDSCs from the HSD group exhibited much higher differentiation into macrophages (F4/80⁺CD11b⁺ cells) than those from the NSD group, while DC differentiation was almost the same between groups (Supplementary Fig. 9a, b). Meanwhile, the high-salt conditions used in the experiment did not influence cell viability (Supplementary Fig. 9c, d). Next, we profiled the mRNA expression of purified tumour-infiltrating M-MDSCs from the NSD or HSD group using Agilent's mouse gene chip array. As shown in Fig. 4c, we identified 5391 differentially expressed genes (DEGs) comprising 2586 upregulated and 2805 downregulated genes, with a fold change greater than 1.5 between M-MDSCs from the NSD and HSD groups. Subsequently, the results demonstrated a significant shift towards an antitumour pro-inflammatory phenotype in HSD-fed mice with a dominant increase in NOS2, TNF, IL-12a, Toll-like receptor 4 (TLR4) and chemokine (C–C motif) ligand 19 (CCL19) expression, and a decrease in lymphocyte antigen 6 complex, locus C1 (Ly6C1), C–C–C motif chemokine ligand 12 (CXCL12), indoleamine 2,3-dioxygenase 1 (IDO1) and CCL2 expression (Fig. 4d). Among these differences, the difference in the expression of IL-12, TNF-α, NOS2, IL-10 and arginase 1 (Arg1) was confirmed by ELISA or qRT-PCR (Fig. 4e, f). Furthermore, immunofluorescent staining also demonstrated significantly increased IL-12 expression and decreased IL-10 expression in the tumour tissues of HSD-fed mice. Moreover, the number of F4/80-positive macrophages in tumour tissues from the HSD group was remarkably higher than that in tumour tissues from the NSD group (Fig. 4g, h). In addition, immunofluorescence staining showed that the blood vessels in tumour tissues were not significantly affected by HSD (Fig. 4g, h). Together, these results demonstrated that HSD effectively induced M-MDSC differentiation and polarised M-MDSCs towards an antitumour and pro-inflammatory phenotype.

Although HSD did not alter the number of PMN-MDSCs in vivo, we asked whether HSD could also influence their function by switching them from a pro-tumour to an antitumour phenotype. After PMN-MDSCs were isolated from NSD and HSD tumour tissues, we investigated the effects of HSD on the immunosuppressive function of PMN-MDSCs. The T-cell proliferation assay indicated that PMN-MDSCs isolated from the HSD group lost their suppressive function and even enhanced T-cell proliferation (Supplementary Fig. 10a). We further examined the effects of HSD on ROS production, one of the main antitumour markers in PMN-MDSCs. ROS production was higher in PMN-MDSCs from the HSD group than in those from the NSD group (Supplementary Fig. 10b). Lastly, after the purified tumour-infiltrating PMN-MDSCs were cultured for 24 h in vitro, cells from the HSD group expressed more TNF-α and ICAM-1 and less prostaglandin E2 (PGE2), Arg1, CCL2 and CCL5 than those from the NSD group (Supplementary

Fig. 10c, d). These data indicated that HSD switched the function of PMN-MDSCs from an immunosuppressive state to a pro-inflammatory and antitumour one.

**HSD enhanced T-cell-mediated antitumour responses.** MDSC accumulation was associated with a reduction in T-cell-mediated antitumour activity during tumour progression[33]. Therefore, we investigated whether promoting MDSC differentiation and inhibiting its immunosuppressive functions by HSD would affect the proliferation and functions of CD4⁺ T and CD8⁺ T cells in tumour animal models. First, we evaluated the total percentages of T cells in the blood, spleen and tumour tissues of the HSD-fed mice compared with those in NSD-fed mice. As shown in Fig. 5a–d, HSD increased the frequencies of CD4⁺ T and CD8⁺ T cells within CD45⁺ cells by ~50% in these tissues of both 4T1 and B16F10 tumour models. Consistent with these data, the proliferation status of CD4⁺ and CD8⁺ T cells within tumours was greater in HSD-treated animals than in NSD-treated animals (Fig. 5e, f, Supplementary Fig. 11a–d). We further showed that HSD treatment resulted in an augmented frequency of IFN-γ-producing CD4⁺ and CD8⁺ T cells in the spleen and tumour of treated mice (Fig. 5g–i, Supplement Fig. 12a, b), indicating a highly activated state of these intratumoural T cells. Meanwhile, HSD significantly reduced the proportion of tumour-induced Tregs in the spleens and tumours of the animal models (Fig. 5g, h, Supplementary Fig. 12c). Studies have recently demonstrated that a high-salt environment could promote the generation of pathogenic Th17 cells with upregulated TNF-α production. Therefore, we investigated whether HSD affected Th17 cell differentiation and the related cytokine expression, and found that HSD enhanced the number of Th17 cells and induced TNF-α expression in tumour tissues (Fig. 5j, Supplementary Fig. 12d). Furthermore, to further confirm the contribution of T cells to the antitumour effect of HSD, we established the two tumour models in the BABL/C-*nu/nu* mice[34,35]. As these mice lack sufficient T-cell-mediated immune reactions, we found no differences in tumour growth indexes (tumour weight and size) between the NSD and HSD groups in the BABL/C-*nu/nu* mice (Supplementary Fig. 13a–f), further demonstrating that the T-cell-mediated antitumour response was key to the HSD-induced antitumour activity.

**HSD enhanced the antitumour activation of PD-1 inhibition.** MDSCs are a highly immunosuppressive population of IMCs that contribute to tumour immune escape by inhibiting antitumour T-cell activity, thereby reducing the efficiency and efficacy of immunotherapies that involve the activation of T cells[36,37]. Therefore, we hypothesised that HSD would impair MDSC function and enhance T-cell-mediated tumouricidal effects,

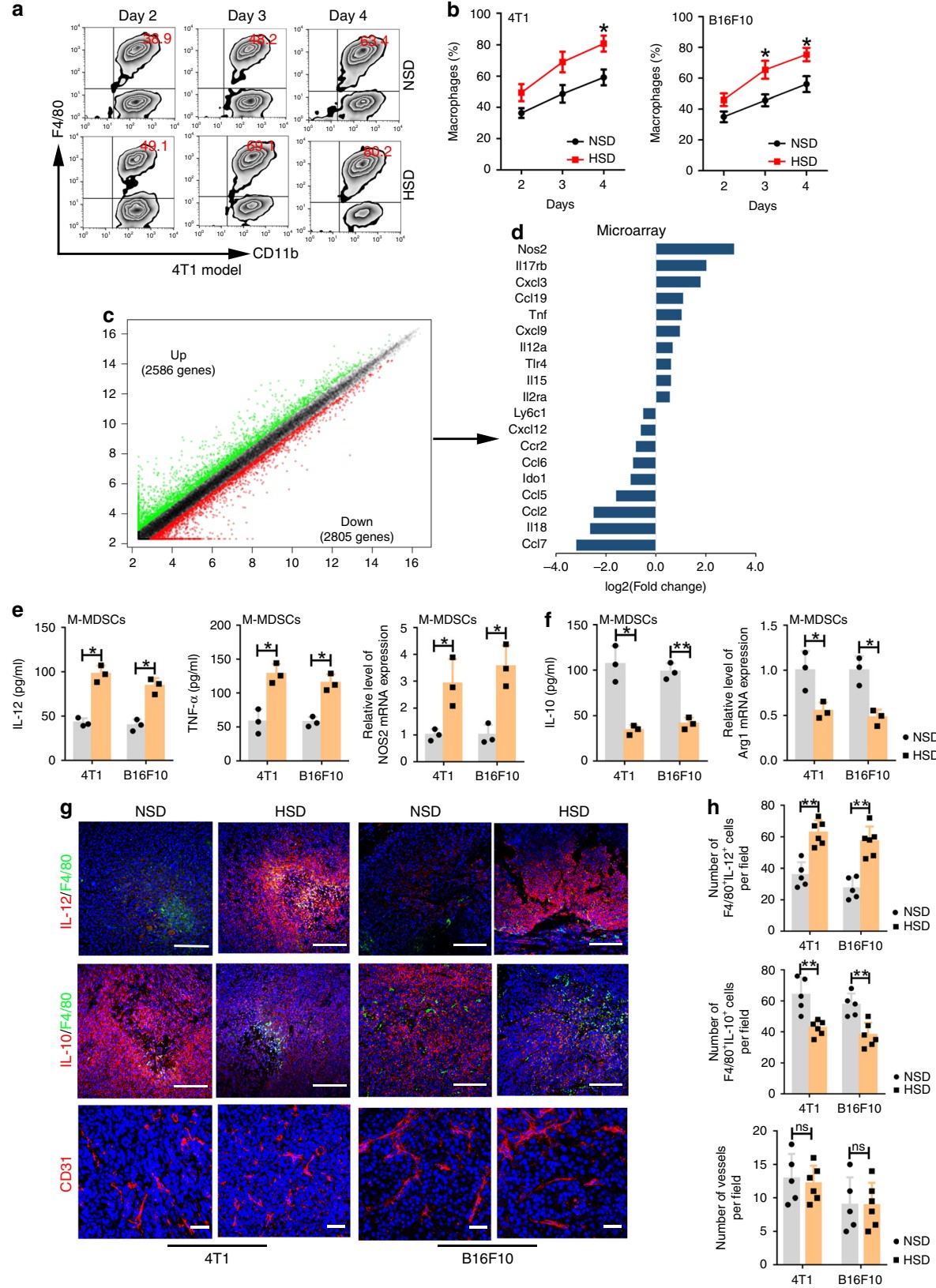

improving the effectiveness of immune checkpoint therapies in cancer treatment. To prove this hypothesis, we examined the effects of HSD treatment in combination with a mouse checkpoint inhibitor, an anti-PD-1 antibody, in 4T1 and B16F10 tumour models. As shown in Fig. 6a–c, treatment with HSD or

the anti-PD-1 antibody alone resulted in a marked retardation of tumour growth across the studied models. However, the combination of HSD and the anti-PD-1 antibody achieved the highest tumour-suppressive response in both the 4T1 and B16F10 model. Furthermore, the combination of HSD and the anti-PD-1

**Fig. 4 HSD promoted MDSC differentiation and functional transformation. a, b** M-MDSCs purified in tumour tissues from the NSD or HSD group were cultured in RPMI-1640 medium containing 10% FBS and 10 ng ml$^{-1}$ GM-CSF for 2, 3 or 4 days. Population of CD11b$^+$F4/80$^+$ cells was evaluated by flow cytometry. One-way ANOVA with post hoc Bonferroni correction; $n = 10$ mice per group; $*p < 0.05$. Bars are expressed as the means ± SEM of three replicates. **c, d** mRNA expression was examined using Agilent Mouse GE v2 Microarrays (4 × 44 k format) after tumour-infiltrating M-MDSCs were isolated from the NSD or HSD group. DEGs in the chip analysis and mRNA expression levels of representative HSD-regulated genes involved in MDSC function from the HSD group; $n = 5$ for the NSD group and 6 mice for the HSD group. **e, f** Tumour-infiltrating M-MDSCs from the NSD or HSD group were cultured in RPMI-1640 medium containing 10% FBS for 24 h. The concentrations of IL-12, TNF-α and IL-10 in the supernatant were tested by ELISA, and NOS2 and Arg1 mRNA expression levels were examined by qRT-PCR. Two-tailed Student's $t$ test; $n = 5$ for NSD and six mice for the HSD group; $*p < 0.05$ and $**p < 0.01$. Bars are expressed as the mean ± SEM of three independent replicates with isolated cells pooled from the same group of mice. **g, h** The effects of HSD on macrophage infiltration, IL-12 and IL-10 levels and angiogenesis (CD31) in tumour tissues were evaluated by immunofluorescent imaging. Green, F4/80; blue, DAPI nuclear staining; red, IL-12, IL-10 or CD31; scale bar, 100 μm. The number of F4/80$^+$IL-12$^+$ cells, F4/80$^+$IL-10$^+$ cells and vessels (CD31$^+$ cells) in high-power optic (for F4/80$^+$IL-12$^+$ cells, F4/80$^+$IL-10$^+$ cells, ×200 magnification; for vessels (CD31$^+$ cells), ×400 magnification) field in stained sections (5 fields for each section; $n = 5$ individual tumours in the NSD group and $n = 6$ individual tumours in the HSD group). The two-tailed Wilcoxon rank-sum tests; $n = 5$ for the NSD group and 6 mice for the HSD group; ns not significant; $**p < 0.01$. Bars are expressed as the mean ± SEM. $n = 5$ data points for the NSD group and 6 data points for the HSD group (the average number of five fields for each section). For all panels, experiments were repeated twice. Source data are provided as a Source Data file.

antibody significantly increased infiltration of T cells in tumour tissues (Fig. 6d–f). The data suggested that HSD is a promising strategy to enhance the efficacy of PD-1 immune checkpoint inhibitors in cancer treatment.

**HSD-regulated p38/MAPK–NFAT5 signalling.** We sought to explain the mechanism by which HSD regulates the differentiation and function of M-MDSCs. First, we analysed cellular processes enriched in DEGs in M-MDSCs isolated from tumour tissues from HSD-fed mice. Interestingly, the 2586 DEGs upregulated (fold change > 1.5) in M-MDSCs from the HSD group were markedly enriched in processes involving cytokine production, cell chemotaxis, IFN-γ production, cell maturation, osmotic stress responses and acute and inflammatory responses (Figs. 4c and 7a). The DEGs related to osmotic stress responses suggested that osmotic stress might be the predominant cellular process represented in the M-MDSCs from the HSD group. From the chip analysis, we found that HSD could also activate the c-Jun N-terminal kinase (JNK) and nuclear factor (NF)-κB pathway (Fig. 7a), and recent studies have suggested that hypertonic stress in mammals is sensed by p38 mitogen-activated protein kinase (p38/MAPK)[38,39]. Therefore, we investigated whether salt influences these pathways in MDSCs in vitro by increasing the concentration of NaCl in the media (an additional 40 mM, referred to as high-salt conditions according to previous reports[40–42]). As shown in Fig. 7b and Supplementary Fig. 14a, b, activation in the presence of this additional 40 mM NaCl for 1 h augmented the phosphorylation of p38/MAPK, but had no effect on JNK phosphorylation or NF-κB activation in M-MDSCs. Meanwhile, high salt promoted the differentiation of M-MDSCs into macrophages, as well as increases M1 marker (TNF-α, IL-12 and NOS2) expression and decreased M2 marker (IL-10 and Arg1) expression. Silencing of p38 blunted this effect of the high-salt conditions (Supplementary Fig. 15a–c). Surprisingly, high-salt stimulation for 24 h, upregulated JNK phosphorylation and NF-κB activation in M-MDSCs, but p38 silencing blunted JNK phosphorylation and NF-κB activation (Fig. 7c, d), suggesting that high salt indirectly regulated JNK and NF-κB signalling cascades via the p38/MAPK pathway. In conclusion, HSD might boost MDSC differentiation and function via the p38/MAPK pathway.

A key transcription factor (TF) involved in the hypertonic stress-induced p38/MAPK cascade is the osmosensitive TF NFAT5[43,44]. Analysis of the microarray dataset indicated that the expression of two key TFs (activating TF 2 [ATF2] and NFAT5) responsible for osmotic stress was enhanced in the

M-MDSCs from the HSD group (Fig. 7e). We further verified the expression of ATF2 and NFAT5 in M-MDSCs from the HSD and NSD groups in the two tumour models using qRT-PCR (Fig. 7f) and western blotting (Fig. 7g), and found that NFAT5 expression was significantly upregulated in M-MDSCs from the HSD group, but that there was no difference in ATF2 expression between the two groups in both tumour models. Lastly, we explored whether high salt (NaCl) also induced NFAT5 expression in vitro. Culturing in the presence of 40 mM NaCl for 24 h markedly upregulated the expression level of NFAT5 (Fig. 7h), whereas p38 deficiency reduced this response (Fig. 7i), suggesting that high-salt conditions led to a p38/MAPK-dependent NFAT5 activation.

As NFAT5 expression could be regulated by p38/MAPK, and was markedly upregulated in the high-salt conditions, we investigated whether this protein played an important role in the high-salt-mediated M-MDSC differentiation and functional transformation using an NFAT5-specific siRNA assay in vitro. The data shown in Fig. 8a, b suggested that there was a significant decrease in the presence of macrophages after silencing of NFAT5 under high-salt conditions. Meanwhile, the level of cytokines (TNF-α and IL-12) and the mRNA expression of NOS2 were downregulated, and IL-10 secretion and Arg1 mRNA expression were upregulated in the NFAT5 siRNA$^+$ NaCl group compared with those in the control siRNA$^+$ NaCl group (Fig. 8c, d). In addition, when M-MDSCs were cultured in high-salt conditions for 24 h, NFAT5 silencing reduced JNK phosphorylation and NF-κB activation (Fig. 8e–g), indicating that NFAT5 indirectly activated the JNK and NF-κB pathway under high-salt conditions. Importantly, western blotting revealed that NFAT5 expression was decreased by the NFAT5-specific siRNA assay (Fig. 8h).

For the in vivo study, we administered a lentivirus carrying NFAT5 siRNA under the Ly-6C promoter (NFAT5 siRNA-LV) or control siRNA-LV intravenously into 4T1 or B16F10 tumour-bearing mice every 2 days for 16 days. Then, we examined the expression of NFAT5 in various immune cell subsets besides M-MDSCs, such as PMN-MDSCs, macrophages and CD3$^+$ T cells. As shown in Supplementary Fig. 16a–e, the lentivirus (NFAT5-siRNA-LV) could silence NFAT5 expression in M-MDSCs and macrophages, but did not affect NFAT5 in PMN-MDSCs and CD3$^+$ T cells at either the mRNA or protein level. In the tumour-bearing mice treated with HSD, NFAT5 silencing in M-MDSCs and macrophages (though adoptively transferred NFAT5-deficient macrophages in vivo could weaken antitumour capacity[26]) could partly restore tumour growth, because PMN-MDSC's function switch affected by HSD could also display

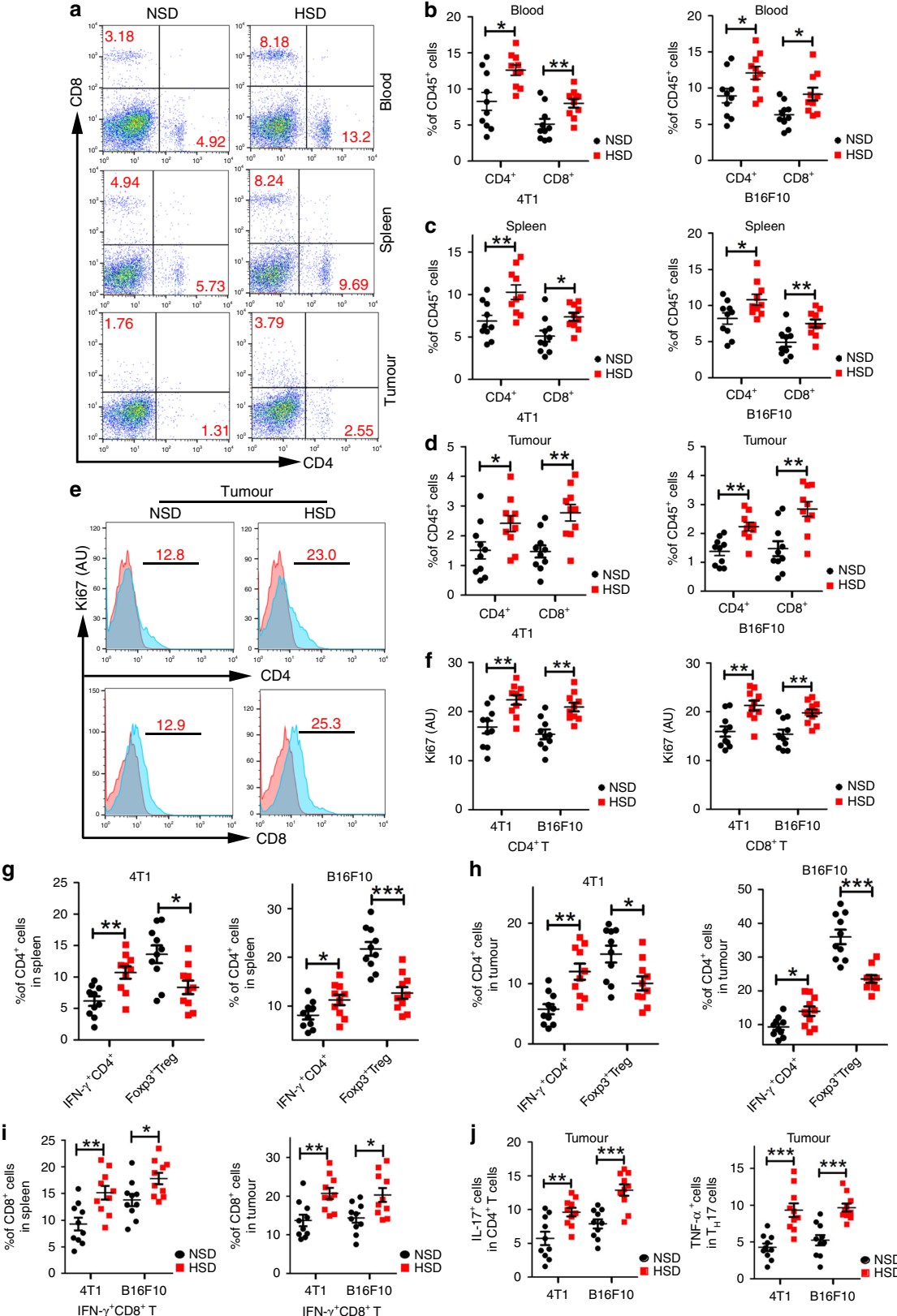

antitumour ability (Supplementary Fig. 17a–c). Meanwhile, the proportion of M-MDSCs and macrophages and M-MDSC functional transformation in tumour tissues was measured. Under high-salt conditions, the frequency of M-MDSCs was downregulated, and an increased proportion of macrophages in tumour tissues was observed. However, knockdown of NFAT5 in

M-MDSCs restored the proportion of M-MDSCs and inhibited the generation of macrophages in vivo (Supplementary Fig. 17d–f). Meanwhile, NFAT5 deficiency suppressed TNF-α and IL-12 secretion and NOS2 expression, and increased IL-10 and Arg1 expression under high-salt conditions (Supplementary Fig. 17g, h).

**Fig. 5 HSD-enhanced antitumour immunosurveillance in vivo. a–d** After NSD or HSD for 16 days, the frequency of CD4$^+$ T cells and CD8$^+$ T cells (right) and representative flow cytometry analysis (left) in the blood, spleen and tumour tissues from 4T1 tumour-bearing mice were assessed within total CD45$^+$ cells. Gating strategy is provided in Supplementary Fig. 21. Data are presented as dot plots extending to minimum and maximum values in one independent experiment, and bars are presented as the mean ± SEM of ten individual mice; the two-tailed Wilcoxon rank-sum tests; $n = 10$ mice per group; *$p < 0.05$ and **$p < 0.01$ vs. the NSD group. **e, f** Analysis of Ki67 expression after gating on CD4$^+$, CD8$^+$ T cells in tumour tissues. The expression of Ki67 in CD4$^+$ T cells and CD8$^+$ T cells (right) and representative flow cytometry analysis (left) in the tumour tissues. Gating strategy is provided in Supplementary Fig. 21. Data are presented as dot plots extending to minimum and maximum values in one independent experiment, and bars are presented as the mean ± SEM of ten individual mice; the two-tailed Wilcoxon rank-sum tests; $n = 10$ mice per group; **$p < 0.01$ vs. the NSD group. **g–j** Splenic T cells and intratumoural T cells from the NSD and HSD groups were further characterised by flow cytometry. Data are presented as dot plots extending to minimum and maximum values in one independent experiment, and bars are presented as the mean ± SEM of ten individual mice; the two-tailed Wilcoxon rank-sum tests; $n = 10$ mice per group. Two-tailed Student's $t$ test; each dot represents one mouse; *$p < 0.05$, **$p < 0.01$ and ***$p < 0.001$ vs. the NSD group. **g, h** Statistical flow cytometry plots showing the proportion of CD4$^+$IFN-γ$^+$ cells and CD4$^+$Foxp3$^+$ Treg cells within total splenic and intratumoural CD4$^+$ T cells. **i** Statistical flow cytometry plots showing the proportion of CD8$^+$IFN-γ$^+$ cells within total splenic and intratumoural CD8$^+$ T cells. **j** Statistical flow cytometry plots showing the proportion of Th17 cells and TNF-α$^+$ Th17 cells within tumour tissues. For all panels, these experiments were repeated twice. Source data are provided as a Source Data file.

We further evaluated whether the antitumour activity of HSD is dependent on promoting MDSC differentiation by MDSC depletion in vivo. Anti-Gr-1 antibody depleted CD11b$^+$ Gr-1$^+$ MDSC in tumour tissues (Supplementary Fig. 18a, b). As shown in Supplementary Fig. 18c–e, MDSC depletion using anti-Gr-1 antibodies could markedly abolish the antitumour activity of HSD, and the combination of HSD and the anti-Gr-1 antibody displayed a similar tumour growth as mice treated with NSD. The results suggested that HSD displayed the antitumour activity by regulating MDSC differentiation and function.

Finally, we isolated normal monocytes (greater than 95% purity, Supplementary Fig. 19a) from the blood of healthy mice to explore whether high-salt conditions influence monocyte differentiation via upregulating NFAT5 expression. The data indicated that high salt enhanced monocyte differentiation into macrophages. However, siRNA-mediated knockdown of NFAT5 in monocytes led to less high-salt-induced macrophage generation (Supplementary Fig. 19b–d).

These results indicated that HSD promoted the differentiation and functional transformation of M-MDSC, even normal monocytes, through p38/MAPK-dependent NFAT5 activation.

## Discussion

HSDs have drawn extensive attention for their negative effects on our health, from worsening of autoimmune disease to increased risk of cardiovascular diseases, obesity, diabetes and fatty liver, with evidence from recent studies[45–47]. On the other hand, new evidence has demonstrated that high-level salt in vivo might contribute to cutaneous antibacterial defences, suggesting that HSD could also be beneficial in skin defence[48]. A recent study also suggests its implication on tumour development[28]. In the present study, we found that HSD resulted in significant accumulation of salt in tumour tissue, modulated MDSC differentiation and function, restored immune surveillanc and eventually retarded tumour growth in mouse cancer models.

The expansion of MDSCs has been widely documented in studies on cancer patients as well as in many animal tumour models[36]. Accumulating evidence has suggested that, upon entering the tumour environment, M-MDSCs quickly differentiate into tumour-associated macrophages (TAMs), leading to enhanced IL-10 secretion and impaired T-cell response[49,50]. Recent studies have suggested that M-MDSCs are an important source of TAMs accumulated in tumour tissue[51]. Therefore, interference in MDSC differentiation and subsequent functionalisation might be useful in tumour therapy. PIR-B deficiency has been shown to promote MDSC differentiation into M1 macrophages and block tumour growth[23], consistent with our present results that M-MDSCs differentiating into M1 macrophages

retarded tumour growth in HSD-fed animals. In addition, according to several studies, based on their functional phenotypes, PMN-MDSCs can be divided into the N1 subgroup, with pro-inflammatory activities, or N2 subgroup, with immunosuppressive and tumour-promoting effects[24,52,53]. Our results indicated that PMN-MDSCs from tumour-bearing HSD-fed mice exhibited an N1 (rather than N2 in NSD-fed animals) phenotype in the tumour environment, relieving more immunosuppression in the tumour microenvironment. Interestingly, the elimination of Gr-1$^+$ cells by anti-Gr-1 antibody abolished the effect of HSD on tumour growth, suggesting that MDSCs play an essential role in the antitumour activity of HSD, though anti-Gr-1 antibody may also affect IMCs, MDSCs and neutrophils (studies have shown that IMCs differentiate into MDSCs in cancer, and PMN-MDSC are pathologically activated neutrophils[54,55]). Our data also showed that HSD inhibited $T_{reg}$ generation possibly through the function of MDSC, which can be compared with other finding that increasing the NaCl concentration in vivo directly impaired Treg functions and resulted in a Th1-type effector signature[7].

We investigated the mechanism by which HSD promoted M-MDSC differentiation towards macrophages with an antitumour phenotype. Through functional enrichment analysis, we demonstrated that the osmotic stress response was one of the major cell processes represented by the DEGs in M-MDSCs in the HSD group. This phenomenon was consistent with the accumulation of salt found in tumour tissues. Previous studies have found that the p38/MAPK pathway regulates NFAT5 in response to high salt, further mediating cellular protective effects and activation of other signalling pathways including inflammatory responses[6,25,56]. In addition, NFAT5 has been implicated in the pathogenesis of autoimmune diseases, such as RA[27], autoimmune type 1 diabetes (T1D)[57] and EAE[3]. Upregulated expression of NFAT5 was found in the synovia of patients with RA, and NFAT5 deficiency markedly inhibited the progression of an arthritis model[58]. Mechanistic studies suggested that activation of the NFAT5 signal could boost the induction of Th17 cells. Moreover, NFAT5 was likely involved in the regulation of Treg activities. In addition to regulating T cells, NFAT5 has also been suggested to significantly increase the expression of IL-12 in macrophages, strengthening the M1 phenotype of the cells[26], which inspired us to investigate the possibility that HSD could direct the differentiation of M-MDSCs into M1 macrophages in tumours via p38-dependent NFAT5 activation. Our data indicated that under a high-salt environment, the p38/NFAT5 axis boosted JNK phosphorylation and NF-κB activity in vivo and in vitro, which contributes to M-MDSC differentiation towards M1 macrophages. To our knowledge, our work provides the first evidence that HSD retards tumour growth via the p38/NFAT5 axis in MDSCs.

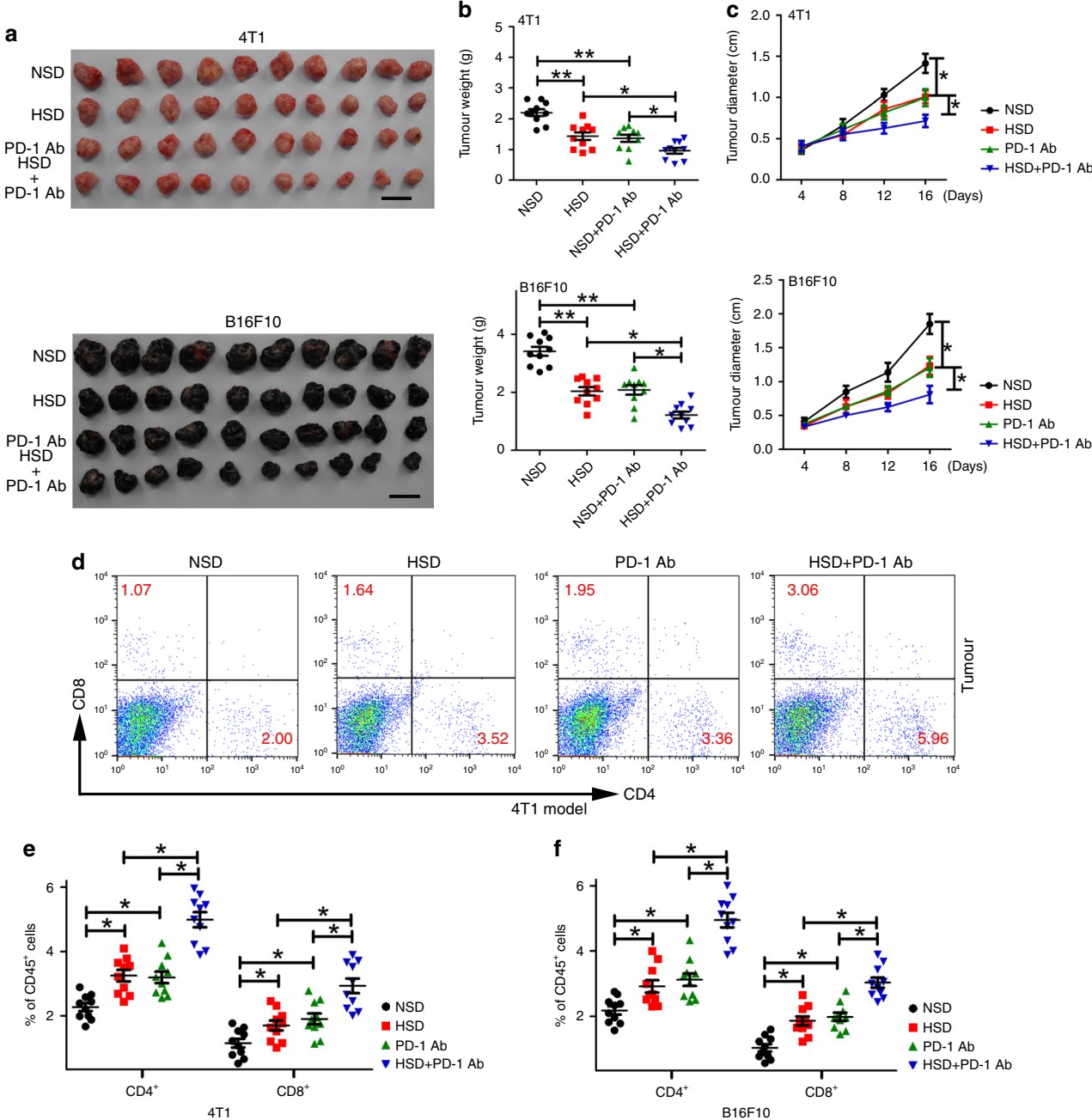

**Fig. 6 Combined treatment with HSD and the anti-PD-1 antibody in tumour-bearing mice. a–c** Female mice were subcutaneously implanted with $1 \times 10^6$ 4T1 or B16F10 cells after starvation for 1 day and randomly assigned to receive the NSD or HSD for 16 days. On days 4, 8 and 12, mice were injected with 4 mg kg$^{-1}$ control IgG antibody or 2 mg kg$^{-1}$ anti-PD-1 via i.p. administration ($n = 10$ for each group). Images of tumours harvested from mice are shown, and tumour size was examined. Scale bar, 1 cm. **d–f** Tumours collected at day 16 were dissociated, and the proportions of CD4+ and CD8+ T cells in the CD45+ cell population were determined by flow cytometry. For all panels, $n = 10$ mice per group. One-way ANOVA with post hoc Bonferroni correction; *$p < 0.05$ and **$p < 0.01$. Data are representative of three independent experiments. Data are presented as dot plots extending to minimum and maximum values in one independent experiment, and bars are presented as the mean ± SEM of ten individual mice; each dot represents one mouse. Source data are provided as a Source Data file.

People in many countries currently consume a diet high in salt, sugar, fat and cholesterol, which has widespread implications in renal, cardiovascular and endocrine homoeostasis, as well as cancer[59,60]. Although some previous studies have shown that salt or salted food is involved in an increased risk of gastric cancer[61], others have suggested little or only weak correlations[60,62]. Furthermore, most conclusions have been obtained from epidemiologic studies. Little is known about the mechanism by which HSD can influence the initiation and promotion of cancer. Recently,

some studies have reported that HSD might stimulate the production of pro-inflammatory cytokines, such as TNF-a and IL-12, and inhibit the expression of IL-10 in vivo[63,64]. Interestingly, these factors could also play a key role in preventing tumour growth by the immune system. This paradox might be due to the different requirements in cancer occurrence and development. Pro-inflammatory factors may contribute to cancer initiation by producing more DNA damage and a higher probability of mutations. After carcinogenesis, cancer development is usually

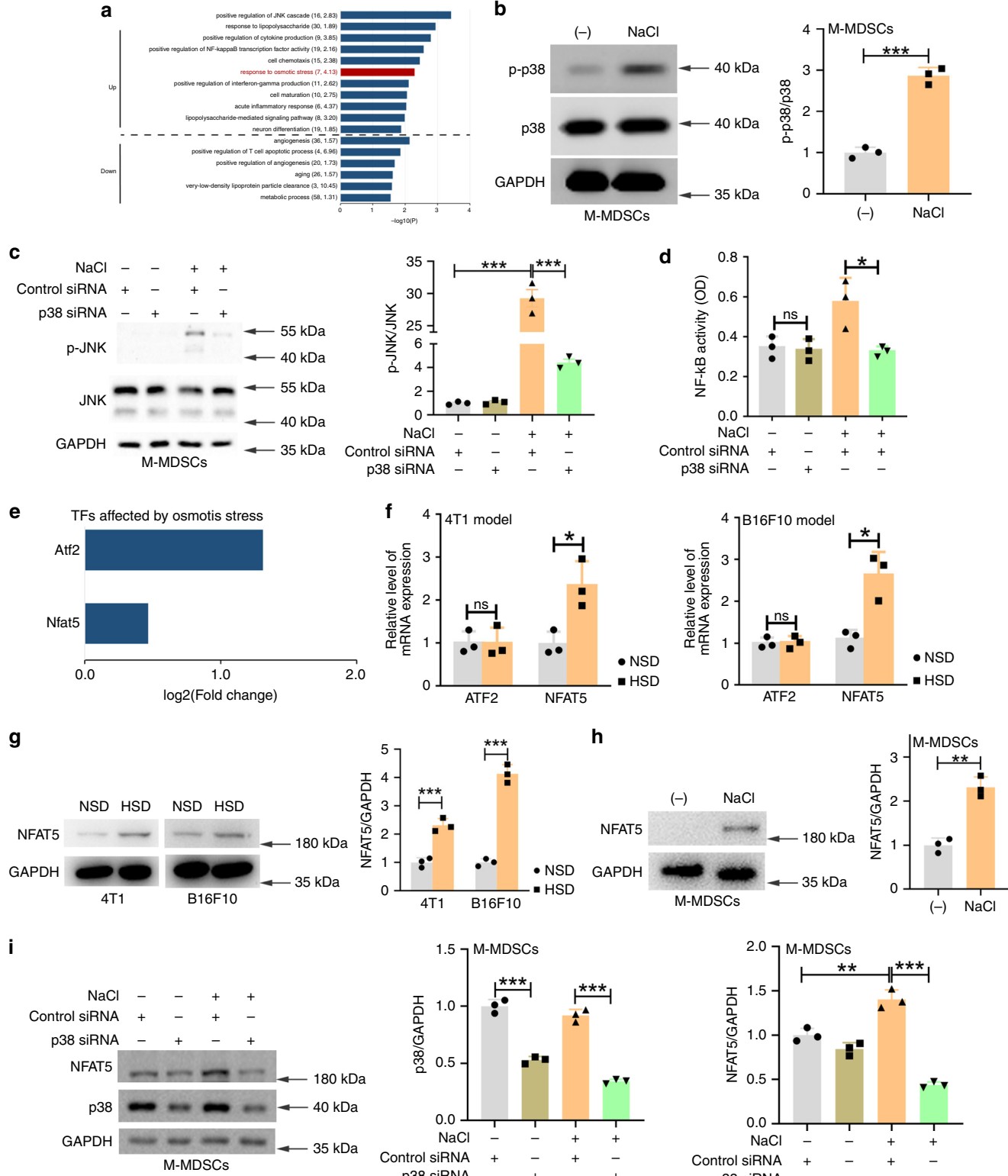

correlated with the immunosuppressive effects conveyed by the tumour cells and tumour microenvironment. In the present study, we used allograft cancer models to simulate a typical cancer developmental process. Based on our findings, the anticancer effect of HSD was derived from its significant reversal of the immunosuppression in cancer models via inhibiting MDSC activity or, possibly, other regulatory effects in other immune cells according to relevant studies.

Cancer immunotherapy represents a breakthrough in medicine. However, limitations exist as only a minority of patients respond to chimeric antigen receptor T-cell therapy or checkpoint inhibitors[65,66]. Mounting evidence has shown that MDSCs represent a key obstacle for efficient cancer immunotherapy[67,68]. Various strategies have therefore been sought to either alter MDSC function (such as cationic polymers[69] or entinostat[37]) or eliminate MDSCs (such as doxorubicin[16]), abolish the

**Fig. 7 HSD promoted the differentiation and function of M-MDSCs by p38/MAPK-dependent NFAT5 expression. a** Significant Gene Ontology (GO) biological process terms associated with M-MDSCs from HSD-fed mice. Cellular processes enriched by the DEGs in M-MDSCs from 4T1 tumour-bearing HSD- or NSD-fed mice. The $p$ value was tested by DAVID software using a hypergeometic test method. $n = 5$ mice for the NSD group and 6 mice for the HSD group. **b** Tumour M-MDSCs were stimulated with an additional 40 mM NaCl for 1 h, and phosphorylated p38 (p-p38) was analysed by western blotting. Right panels: quantification data. **c, d** Tumour M-MDSCs were transfected with p38-specific siRNA or control siRNA for 2 days, and cultured with an additional 40 mM NaCl for 24 h. Phosphorylated JNK (p-JNK) was analysed by western blotting (right panels: quantification data), and NF-κB activity was determined by the p65 subunit DNA-binding ability. **e** Two TFs identified as upregulated DEGs in tumour M-MDSCs in 4T1 tumour-bearing HSD-fed mice. **f** ATF2 and NFAT5 mRNA level in tumour M-MDSCs in 4T1 or B16F10 tumour-bearing HSD- or NSD-fed mice was tested by qRT-PCR. **g** NFAT5 expression in M-MDSCs from tumour-bearing HSD- or NSD-fed mice was analysed by western blotting. Right panels: quantification data. **h** An equal number of tumour M-MDSCs were cultured for 24 h in the absence or presence of an additional 40 mM NaCl. NFAT5 expression was determined by western blotting. Right panels: quantification data. **i** Tumour M-MDSCs were transfected with p38-specific siRNA for 2 days, and cultured in the absence or presence of an additional 40 mM NaCl for 24 h. The level of NFAT5 was examined by western botting. Right panels: quantification data. For **b**–**d** and **f**–**h**, two-tailed Student's $t$ test; ns not significant; $*p < 0.05$, $**p < 0.01$ and $***p < 0.001$; $n = 5$ mice for the NSD group and 6 mice for the HSD group. Bars are expressed as the means ± SEM of 3 independent replicates with isolated cells pooled from the same group of mice. For **i**, $n = 6$ mice. $**p < 0.01$ and $***p < 0.001$. The data are expressed as the mean ± SEM of independent three replicates; one-way ANOVA with post hoc Bonferroni correction. Experiments were repeated three times. Source data are provided as a Source Data file.

immunosuppression, restore immune surveillance and enhance the efficacy of cancer immunotherapy. Here, our data demonstrated that HSD partly relived the immunosuppression of MDSCs to enhance T-cell proliferation and function by promoting MDSC differentiation and changing their function, suggesting that HSD might enhance immunotherapies. Our study also demonstrated that, compared with monotherapy, the combination of HSD plus the anti-PD-1 antibody significantly decreased tumour growth by increasing T-cell infiltration. These findings might help to explain the differences in the response to cancer immunotherapy and benefit cancer immunotherapy by providing evidence supporting the integration of combinatorial therapeutic strategies.

Although our study is intriguing, whether HSD impacts the general health of animals still needs to be investigated. We found no marked changes in spleen, liver or kidney indexes, and no evidence of hepatotoxicity or nephrotoxicity; even body weight was unchanged by 16 days of high-salt intake, although HSD-fed mice consumed markedly more food and water than NSD-fed mice. In addition, our results were mainly based on transplanted tumours, indicating that HSD displays antitumour ability after tumour occurrence, but its role in tumorigenesis has been neglected. Therefore, whether HSD has antitumour activity in tumorigenesis needs to be investigated.

In summary, we demonstrated that HSD can restore antitumour immunosurveillance via regulating MDSC differentiation and function. We further identified a mechanism by which p38-dependent NFAT5 expression, secondary to high-salt exposure, was involved in mediating the differentiation of M-MDSCs into M1-type macrophages. Our findings indicate that HSD has an unexpected potential for immune regulation that may have further implications for cancer immunotherapy.

## Methods

**Cell lines.** B16F10 (TCM36) and 4T1 (TCM32) cell lines were obtained from the Shanghai Cell Bank of Chinese Academy of Sciences (Shanghai, China). All cell lines were analysed for mycoplasma contamination using a mycoplasma stain assay kit (Beyotime Biotechnology, Shanghai, China). All cell lines were cultured in RPMI-1640 or DMEM supplemented with 10% heat-inactivated foetal bovine serum (FBS), 100 U ml⁻¹ penicillin and 0.1 mg ml⁻¹ streptomycin.

**Protein and qRT-PCR.** For western blotting analyses, cells were homogenised in RIPA buffer (Beyotime Biotechnology, Shanghai, China). After the samples were mixed with 2× Laemmli buffer and boiled for 5 min, 50 μg of each protein were subjected to sodium dodecyl sulfate polyacrylamide gel electrophoresis and transferred onto polyvinylidene fluoride membranes (Millipore, Bedford, MA, USA) using Semi-Dry Trans-Blot (Bio-Rad Laboratories). Immunoblots were first incubated in 5% nonfat dry milk for 1 h at room temperature (RT) and then incubated with the indicated primary antibodies at appropriate dilutions overnight at 4 °C. Subsequently, immunoblots were washed with PBS Tween, incubated with horseradish

peroxidase-conjugated anti-rabbit immunoglobulin G (CST, Danvers, MA, USA; 7074 S) for 60 min at RT and detected using an enhanced chemiluminescence system (Millipore, Billerica, MA, USA) by Tanon 4200SF Gel Imaging System (Shanghai, China). The protein bands were quantitated by densitometry using ImageJ software (NIH, Bethesda, MD). Information regarding the antibodies used is listed in Supplementary Table 1. The pre-stained Protein Ladder (26616 or 26634, Thermo Scientific) was used as size standards in western blotting. All blots and gels are accompanied by the locations of molecular weight/size markers, which, together with uncropped and unprocessed scans of the most important blots (Figs. 7h and 8h, Supplementary Figs. 14a and 16b), are provided in combined source data.

For qRT-PCR analyses, total RNA was extracted by TRIzol reagent (Invitrogen, Carlsbad, CA, USA) according to the manufacturer's instructions, and 500 ng of RNA was converted to cDNA using PrimeScript™ RT Master Mix (Perfect Real Time) (Takara, Da Lian, China). qRT-PCR assays to test the differences in mRNA expression were performed on a 7300 Sequence Detection System (Applied Biosystems, Foster City, CA, USA) using EvaGreen Dye (Biotium, Hayward, CA, USA). mRNA expression was normalised to the expression of β-actin. Transcript levels were calculated relative to the levels of the internal control β-actin, and are expressed as $2^{-\triangle CT}$, in which $\triangle CT = C_{T, mRNA} - C_{T, \beta-actin}$. The gene-specific primers used can be found in Supplementary Table 2.

**RNA interference.** Mouse NFAT5 siRNA (sense: 5′-GGUACAGCCUGAAAC CCAATT-3′ and antisense: 5′-UUGGGUUUCAGGCUGUACCTT-3′) was obtained from Gene-Pharm (Shanghai, China). p38 MAPK siRNA (CST#6564) was purchased from Cell Signalling Technology (Danvers, MA, USA). Electroporation was performed using a Gene Pulser electroporation (Bio-Rad, Hercules, USA) and a 0.1-cm electrode gap cuvette (Bio-Rad). The electroporated cells were maintained at 37 °C for 20 min, and then transferred into six-well plates. The treatments were performed 24 h after siRNA transfection.

**Microarray analysis.** The RNA extracts were first evaluated by a Nanodrop ND-1000 spectrophotometer (Thermo Fisher Scientific, Waltham, MA), and RNA quality was determined by the ratios of A260/A280 (close to 2) and A260/A230 (close to 2). Microarray analysis of the genome-wide expression profile was done by KangChen Biotech (Shanghai, China) with the Agilent Mouse GE v2 Microarrays (4 × 44 k format) (Agilent Technologies).

**Animals and tumour models.** Female and male C57BL/6J and BALB/C mice (6–8 weeks old) and female BABL/C-*nu/nu* mice (6–8 weeks old) were obtained from the Animal Centre of Yangzhou University (Yangzhou, China), and the permission number is SCXK(su)2017-007. Mice were maintained under a 12-h light/12-h dark cycle in specific pathogen-free (SPF) conditions at 22–24 °C. All mice were fed normal chow diets (for C57BL/6 J and BALB/C mice, rodent diet, 1010041, Shoobree, Xietong Organism, Jiangsu, China; for BABL/C-*nu/nu* mice, rodent diet, 1010019, Shoobree, Xietong Organism, Jiangsu, China) and water ad libitum, and were treated in strict accordance with the Nanjing University guidelines (Permit NO. 2011-039) and the ARRIVE guidelines (Animal Research: Reporting of In Vivo Experiments).

To generate the heterotopic tumour model, mice were subcutaneously injected with 1 × 10⁶ cells (B16F10 or 4T1) after 1 day of starvation. The cells were injected into the left armpit of each mouse. Then, mice were randomly assigned to receive the NSD (0.4% NaCl in chow plus tap water) or HSD (4% NaCl in chow plus 1% NaCl in the water) via random lottery for 16 days (10 mice per group). We tested the tumour size with callipers and weighed the tumour samples upon harvest. The main ingredients and caloric content in NSD and HSD are listed in Supplementary Table 3.

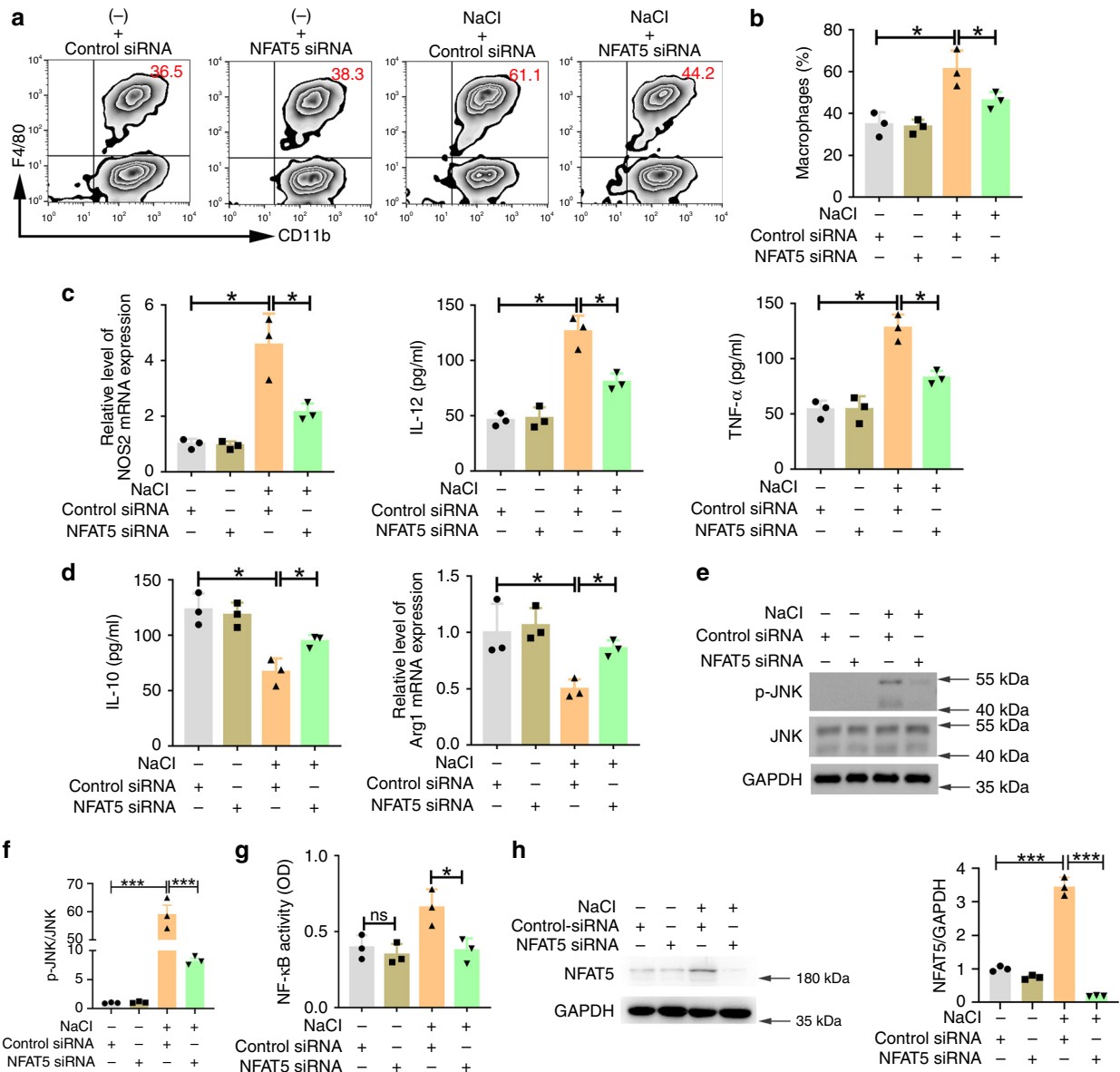

**Fig. 8 NFAT5 silencing inhibited high-salt-mediated M-MDSC differentiation and functional transformation. a**, **b** Purified tumour M-MDSCs were transfected with NFAT5-specific siRNA and cultured in the absence or presence of an additional 40 mM NaCl for 3 days. Populations of CD11b⁺F4/80⁺ cells were tested by flow cytometry. **c**, **d** Purified tumour M-MDSCs were transfected with NFAT5-specific siRNA for 2 days and cultured in the absence or presence of an additional 40 mM NaCl for 24 h. The expression of IL-12, TNF-α and IL-10 in the supernatant was determined by ELISA. NOS2 and Arg1 mRNA expression levels were evaluated by qRT-PCR. **e**–**g** p-JNK was analysed by western blotting; quantification data of western blotting and NF-κB activity were determined by the p65 subunit DNA-binding ability. **h** NFAT5 expression was analysed by western blotting. Right panels: quantification data. For all panels, *$p < 0.05$ and ***$p < 0.001$. Bars are expressed as the means ± SEM of three independent replicates with isolated pooled cells. These experiments were repeated three times and were replicated with similar results; $n = 5$ mice; one-way ANOVA with post hoc Bonferroni correction. Source data are provided as a Source Data file.

For the analysis of the effect of HSD on food and water intake and mouse body weight, mice were first randomly assigned to two groups via random lottery, the tumour-free group and tumour group. Mice in the tumour group were subcutaneously injected with $1 \times 10^6$ cells (B16F10 or 4T1) into the left armpit after 1 day of starvation. Then, mice were randomly assigned to four groups (10 mice per group): NSD + tumour-free group (NSD group), HSD + tumour-free group (HSD group), NSD + tumour group (NSD + tumour group) or HSD + tumour group (HSD + tumour group). Mice received the NSD or HSD for 16 days. On days 4, 8, 12 and 16, the food and water intake of mice were tested, and the changes in the body weights of mice were tested every 4 days.

For the analysis of the effect of HSD on the general health of the animals, mice were randomly assigned to receive an NSD or HSD via random lottery for 16 days (10 mice per group). At day 16, different tissues were harvested and weighted. After the blood samples clotted for 1 h at RT, they were centrifuged at 835×g for 15 min for serum collection. Animal serum ALT, AST, BUN and creatinine levels were

examined for the evaluation of liver and kidney function, according to the manufacturer's instructions. The ALT assay kit (C009-2), AST assay kit (C010-2), urea assay kit (C013-2) and creatinine assay kit (sarcosine oxidase, C011-2) were purchased from Nanjing Jiancheng Bioengineering Institute (Nanjing, China).

For the analysis of the effect of the combination of HSD and anti-PD-1 treatment in the subcutaneously implanted tumour models, mice were subcutaneously injected with $1 \times 10^6$ cells (B16F10 or 4T1) into the left armpit after 1 day of starvation, and then randomly assigned to four groups (10 mice per group) via random lottery: the NSD group, HSD group, anti-PD-1 antibody group or combined HSD and anti-PD-1 antibody groups. Mice were fed the NSD or HSD for 16 days. On days 4, 8 and 12, mice were injected with 4 mg kg⁻¹ isotype controls (clone 2A3, Bioxell, West Lebanon, NH) or 2 mg kg⁻¹ anti-PD-1 antibody (clone RMP1-14, Bioxcell) via intraperitoneal (i.p.) administration. At day 16, we measured tumour size with callipers and weighed the tumour samples upon harvest.

For the analysis of the effect of the combination of HSD and angiotensin-converting enzyme inhibitor (captopril) treatment in the subcutaneous implantation tumour model, mice were subcutaneously injected with $1 \times 10^6$ cells (B16F10 or 4T1) into the left armpit after 1 day of starvation. Then, mice were randomly assigned to four groups (6 mice per group) via random lottery: the NSD group, HSD group, captopril group or combined HSD and captopril groups. Mice were fed the NSD or HSD for 16 days. Meanwhile, mice in the captopril group received captopril (10 mg kg$^{-1}$ d$^{-1}$) by gavage every day. At day 16, we measured tumour size with callipers and weighed the tumour samples upon harvest. Captopril was purchased from Dalian Meilun Biotech Co., Ltd. (Dalian, China).

For the analysis of the effect of NFAT5 on HSD-mediated differentiation of M-MDSCs into macrophages in vivo, mice were subcutaneously injected with $1 \times 10^6$ cells (B16F10 or 4T1) into the left armpit after 1 day of starvation and randomly assigned to four groups (6 mice per group) via random lottery: the NSD group, HSD group, NFAT5 siRNA-LV group or the combined HSD and NFAT5 siRNA-LV groups. Mice were fed the NSD or HSD for 16 days. Meanwhile, mice were administered with lentivirus carrying NFAT5 siRNA under the Ly-6C promoter (NFAT5 siRNA-LV, GenePharma Company, Shanghai, China) via the tail vein ($10^9$ PFU per mice) every 2 days. At day 16, we measured tumour size with callipers and weighed the tumour samples upon harvest.

For the analysis of the effect of HSD after MDSC depletion in the subcutaneously implanted tumour models, mice were subcutaneously injected with $1 \times 10^6$ cells (B16F10 or 4T1) into the left armpit after 1 day of starvation and then randomly assigned to three groups (10 mice per group) via random lottery: the NSD group, HSD group or combined HSD and anti-Gr-1 antibody groups. Mice were fed the NSD or HSD for 16 days. Mice were injected with 200 μg of isotype controls (clone LTF-2, Bioxell, West Lebanon, NH) or 200 μg of anti-Gr-1 monoclonal antibody (clone RB6.8C5, Bioxcell) via intraperitoneal (i.p.) administration every 2 days. At day 16, we measured tumour size with callipers and weighed the tumour samples upon harvest.

All in vivo experiments were conducted by using a double-blind manner so that the group identity for each tumour-bearing mice was not disclosed to the scientists until all results were completed. All mice were adapted to the laboratory for 7 days to observe the health status. The mice were weighed when used in experiments (Supplementary Fig. 2c), and mice could be excluded from the experiment with rapid weight loss or infection during the progression of the experiment. Mice were killed via cervical dislocation after completing all animal experiments.

**Histological and immunofluorescence analysis.** After tumour tissues were fixed with 4% paraformaldehyde, they were embedded in paraffin and sectioned. The sections were de-paraffinised in xylene and rehydrated in graded alcohol. For histologic analyses, the sections were counterstained with haematoxylin and eosin (H&E), and the tumour necrotic area per field was quantified using ImageJ software (NIH). For immunofluorescence analyses for tumour tissues, after antigen retrieval by boiling sections in citrate antigen retrieval solution (Beyotime Biotechnology, Shanghai, China) for 10 min, the sections were blocked with 5% bovine serum albumin (BSA) for 1 h and incubated with primary antibodies at 4 °C overnight. The slices were washed and incubated for 1 h with fluorescence-conjugated secondary antibodies. For immunofluorescent analyses for T cells, isolated CD4$^+$ T cells and CD8$^+$ T cells were fixed with 4% paraformaldehyde for 15 min, penetrated with 0.3% Triton X-100 for 10 min, blocked with 3% BSA for 30 min and stained with primary antibodies for 2 h at RT. Cells were washed and incubated for 1 h with fluorescence-conjugated secondary antibodies. Finally, the nuclei were stained with DAPI (Beyotime Biotechnology, Shanghai, China) and coverslipped. Sections were examined using a Nikon confocal microscope, and each immunofluorescent stain was repeated three times using serial sections. The number of double-positive cells and vessels per field was quantified using ImageJ software. The quantitation of double-positive cells and vessels per field was conducted by using a single-blind manner so that the section identity for different groups was not disclosed to the scientists until all results were completed. Information regarding the antibodies used is listed in Supplementary Table 1.

**Cell flow cytometry.** Cell suspensions from the blood, spleen or tumour tissues were filtered through Nylon cell strainers (70 μM, Falcon, USA), and red blood cells (RBCs) were lysed. After cells were washed with phosphate-buffered saline (PBS) containing 1% BSA, cells were blocked with 1% BSA at 4 °C for 30 min. For extracellular staining, 7-AAD was used for live/dead cell determination. For intracellular staining, Fixability viability Dye 520 (FVD eFluor 520, eBioscience) was used for live/dead cell determination. For extracellular staining, $1 \times 10^6$ cells were incubated with antibodies for 30 min on ice and then washed with 1% BSA. For intracellular staining, cells were first stained on ice for 30 min with surface-staining antibodies, washed, fixated and permeabilised with the FIX&PERM Kit (Multi Sciences, GAS003) and stained with cytokine antibodies. For analysis of Tregs, Tregs were assayed by the Mouse Regulatory T cell Staining Kit (Multi Sciences, KTR201-100) according to the manufacturer's recommendation. The samples were resuspended in 500 μl of 1% BSA and analysed by flow cytometry. Flow cytometry was performed on a FACSCalibur device (Becton-Dickinson, Mountain View, CA, USA), and the data were evaluated with FlowJo v10.0 software (Tree Star). The proportions of immune cells examined by flow cytometry were conducted by using a single-blind manner so that the flow cytometry data identity for different groups was not disclosed to the scientists until all results were completed. Information regarding the antibodies used is listed in Supplementary Table 4.

**Cytokine antibody array and ELISA.** Tumour-bearing mice were fed the HSD or NSD for 16 days. Tumour tissue was then homogenised in PBS containing protease inhibitors. After homogenisation, Triton X-100 was added to the tumour homogenates (at a final concentration of 1%), which were then thawed, frozen at −80 °C for 1 day and centrifuged at 13,362×g for 30 min at 4 °C to remove cellular debris. After protein quantitation using a BCA protein assay kit, 400 μg of tumour homogenates from both the NSD and HSD groups were used. Blood samples were obtained from mice in each cohort and allowed to clot for 1 h at RT before centrifuging for 15 min at 835×g rpm. Serum samples were harvested and mixed at equal volumes. The mixed homogenates and serum were assayed by the Proteome Profiler Mouse XL Cytokine Array (ARY028, R&D Systems, Inc.) according to the manufacturer's recommendation. Briefly, the membranes were blocked with the blocking buffer (Array Buffer) at RT for 1 h and incubated with the samples (200 μl of serum or 200 μg of tumour lysates) at 4 °C overnight. Membranes were washed three times with Wash Buffer at RT for 10 min per wash and incubated with biotin-conjugated antibodies at RT for 1 h. Next, the membranes were washed and incubated with horseradish peroxidase-conjugated streptavidin at RT for 30 min. After the membranes were washed three times with wash buffer, the membranes were incubated with detection buffer for 1 min. We used a luminescence detector (Chemi Scope 6300, CSI, Shanghai, China) for detection, and the results were digitised and subjected to image analysis (ImageJ, freely downloaded from the National Institutes of Health website, http://rsbweb.nih.gov/ij/). We obtained the relative protein concentrations by subtracting the background staining and normalising to the positive controls on the same membrane. The complete list of multiple cytokines, chemokines, growth factors and other soluble proteins in Proteome Profiler Mouse XL Cytokine Array is displayed in Supplementary Table 5.

For ELISA, the TNF-α, IL-12, INF-γ, ICAM-1, IL-6, GM-CSF, IL-10 and PGE2 concentration in serum, tissue lysates or cell supernatant was tested by using ELISA kits (4A Biotech Co., Ltd., Beijing, China), according to the manufacturer's instructions.

**MDSC isolation.** Total MDSCs (T-MDSCs, CD11b$^+$Gr-1$^+$), M-MDSCs (CD11b$^+$Ly6-C$^+$) and PMN-MDSCs (CD11b$^+$Ly6-G$^+$) were isolated from the tumour tissues by the Myeloid-Derived Suppressor Cell Isolation Kit according to the manufacturer's instructions (Miltenyi Biotec, Bergisch Gladbach, Germany), with all steps performed at 4 °C. Briefly, tumour-bearing mice fed the NSD or HSD for 16 days were sacrificed, and tumour tissues were harvested. Fresh tumour tissues were cut into pieces and then enzymatically digested with 0.2% collagenase IV (weight per volume) and 0.1% DNase (wt per vol) (Sangon Biotech, China) for 1 h at 37 °C. After the cells were filtered through 70 μM Nylon cell strainers (Falcon, USA), the resulting cell suspension was centrifuged at 93×g for 5 min. The cells were depleted of RBCs using RBC lysis buffer and washed twice with cold PBS containing 1% BSA. After treatment with FcR blocking reagent (50 μl per $10^8$ cells) for 10 min, cells were stained with the biotin-conjugated granulocyte receptor (GR)-1 or Ly-6G monoclonal antibody and further labelled with anti-biotin or streptavidin microbeads. Then, cells were passed through the MACS column (MS or LS separation column) for magnetic cell separation. The retained cells were M-MDSCs or PMN-MDSCs, and the purity and cell viability of the MDSC subfractions were typically greater than 90% (Supplementary Fig. 20a, b).

**Macrophage, T-cell isolation from tumour tissues.** Fresh tumour tissues were prepared as single-cell suspensions, minced in the DMEM with 0.2% collagenase IV (weight per volume) and 0.1% DNase (wt per vol) for 1 h at 37 °C and filtered through 70 μM Nylon cell strainers (Falcon, USA). F4/80$^+$ cells, CD4$^+$ T cells and CD8$^+$ T cells were isolated from single-cell suspensions by anti-F4/80 microbeads ultrapure mouse (130-110-443, Miltenyi Biotec), Anti-Mouse CD4—DM (551539, BD) and Anti-Mouse CD8a Particles—DM (551516, BD) according to the manufacturer's instructions, with all steps performed at 4 °C.

**M-MDSC and monocyte differentiation.** For M-MDSC differentiation experiments, freshly isolated tumour M-MDSCs from tumour tissues from the NSD and HSD groups were cultured in RPMI-1640 medium supplemented with 10% FBS and 10 ng ml$^{-1}$ GM-CSF (Peprotech) for 2, 3 and 4 days. For normal monocyte differentiation, monocytes were isolated from blood using a mouse peripheral monocyte separation medium kit according to the manufacturer's protocol (TBD, TBM2011M). Isolated monocytes were cultured in RPMI-1640 medium supplemented with 10% FBS, 10 ng ml$^{-1}$ GM-CSF (Peprotech) and an additional 40 mM of NaCl for 3 days. CD11b$^+$F4/80$^+$ cells were analysed by flow cytometry.

**Cell viability assay.** During M-MDSC differentiation, $1 \times 10^6$ cells were collected and resuspended in 500 μl of 1% BSA. Then, cells were stained with 5 μl of 7-aminoactinomycin-D (7-AAD). After 10 min, cells were analysed by flow cytometry.

**T-cell proliferation assay**. Total spleen cells from healthy mice were stained with carboxyfluorescein succinimidyl ester (CFSE, Beyotime) at 2.5 μM and cultured with PMN-MDSCs isolated from tumour tissues from different groups in a 96-well plate. Spleen cells were co-cultured with PMN-MDSCs at a 2:1 ratio. These cells were stimulated with Con A (2 μg ml$^{-1}$, Sigma-Aldrich, New Jersey, USA) and left in culture for 3 days before being analysed by flow cytometry.

**ROS detection**. PMN-MDSCs isolated from tumour tissues from NSD- or HSD-fed mice were incubated with dichlorofluorescein diacetate (DCFH-DA, ROS Assay Kit, Beyotime Biotechnology, Shanghai, China) for 30 min at 37 °C. Cells were then stained with anti-Ly-6G antibodies. The level of ROS was tested using flow cytometry as described.

**NF-κB (p65) activity assay**. Nuclear extracts from M-MDSCs were prepared using a nuclear extract kit (Cayman Chemical, #10009277, Ann Arbor, MI, USA). The nuclear extracts were adjusted to the same protein concentration by a BCA protein assay kit, and then the DNA-binding activity of NF-κB was examined using a NF-κB (p65) TF assay kit (Cayman Chemical, #10007889) according to the manufacturer's instructions.

**Chemical analysis of tissue electrolyte and water content**. Thymus, liver, spleen, heart, lung and tumour tissues isolated from the NSD and HSD groups were weighed (wet weight), frozen at −80 °C for 24 h and then freeze-dried using a Labconco FreeZone 2.5 L freeze dryer (Table Model, USA) for 48 h (dry weight [DW]). The difference between the wet weight and DW was tissue water content. We then put the tissues in 50% HNO$_3$ for 48 h and incubated them at 190 °C for 12 h. Finally, we dissolved the tissues in 1% HNO$_3$ and examined Na$^+$ and K$^+$ content by atomic adsorption spectrometry (HITACHI180-80) and Cl$^-$ content by titration with 0.1 N silver nitrate (Model Titrando, German Metrohm).

**X-ray fluorescence spectrometry analysis**. Tumours isolated from the NSD and HSD groups were frozen at −80 °C for 24 h and then desiccated in a freeze dryer for 48 h. The Na$^+$, K$^+$ and Cl$^-$ content of tumour samples was assayed by X-ray fluorescence spectrometry at the Center of Modern Analysis (Nanjing University).

**Osmolality measurements**. Tissue osmolality was examined by using a vapour-pressure osmometer (Vapro 5520, Wescor, Logan, UT) as previously reported[43]. Briefly, tissue obtained from anaesthetised NSD- or HSD-fed mice was placed in a 1.5-ml tube and snap-frozen by dry ice. Tissue fluid from frozen tissue that had been fragmented was added into filter discs and osmolality was measured. The whole-blood osmolality measurements were identical to those of serum, which was measured by a STY-1A pressure osmometer (TDTF, Tianjing, China).

**Statistical analysis**. All data are presented as the mean ± standard error. Data visualisation and statistical analysis were performed using Prism 5 or 7 (GraphPad Software Inc., La Jolla, CA, USA), after all the data sets for normal distribution were tested by IBM SPSS Statistics 20 software (IBM Corp., Armonk, NY, USA). Differences between two groups were evaluated using the two-tailed Student's $t$ test or the two-tailed Wilcoxon rank-sum tests. Differences between multiple groups were compared using one-way ANOVA with post hoc Bonferroni correction or one-way ANOVA with Dunnett's test. Details of statistical analyses including sample numbers ($n$) are included in the respective figure legends. NS indicates not statistically significant, and the difference was considered significant when $p < 0.05$. At last, the results and relevant accuracy about all in vivo experiments are provided in figures and figure legends.

**Reporting summary**. Further information on research design is available in the Nature Research Reporting Summary linked to this article.

## Data availability

All relevant data are available in the article, Supplementary Information or from the corresponding author upon request. The source data underlying all figures are provided as a Source Data file. The microarray data have been deposited in the Gene Expression Omnibus (GEO)/NCBI public database (accession no. GSE125430, https://www.ncbi.nlm.nih.gov/geo/query/acc.cgi?acc=GSE125430).

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

## Acknowledgements

This study was funded by the National Key Research and Development Programme of China (2017YFC0909702), the National Natural Science Foundation of China (81973273, 81673380, 31971309 and 31671031), the Jiangsu Province Funds for Distinguished Young Scientists (BK20170015) and the Fundamental Research Funds for the Central Universities (020814380115). C.W. acknowledges the funding support from the Science and Technology Development Fund, Macau SAR (FDCT No. 080/2016/A2, 0018/2019/AFJ, 0097/2019/A2) and the University of Macau (MYRG2017-00028-ICMS). L.D. acknowledges the UM Macau Distinguished Visiting Scholar (MDS) Programme. This study was also supported by the funds for the International Cooperation and Exchange of the Natural Science Foundation of China and the Science and Technology Development Fund (31961160701), Anhui provincial Natural Science Foundation (1908085QC131) and Grants for Scientific Research of BSKY (XJ201726) from Anhui Medical University.

## Author contributions

W.H., L.D., C.W. and J.Z. conceived and designed the experiments; W.H., J.X., R.M. and Q.L. performed the experiments; W.H., J.X., D.L. and H.Z. analysed and interpreted the data; W.H., L.D., C.W. and J.Z. contributed to the preparation of the paper.

## Competing interests

The authors declare no competing interests.
