## [Peer Review File · Nature Communications]

Reviewers' comments:

Reviewer #1:cancer immunology

(Remarks to the Author):

The study by He et al., 'High-salt diets inhibit tumor growth in mice via regulating myeloid-derived suppressor cells (MDSCs) differentiation', presents new and interesting data that solid tumor growth in allogeneic tumor transplant models is inhibited and delayed by high-sodium diets in mice. The authors claim that this effect is mediated through effects on myeloid derived suppressor cells and other immune cells, involving the induction of NFAT5. Although the presented data is of interest, the study in its current form seems to be highly premature and the presented data pieces are rather descriptive and loosely connected, lacking sound mechanistic data, raising critical doubts that the described phenomena is based on the proposed pathway. In addition, some of the presented data needs further clarification and it is unclear if experiments include replicates.

Major points:

1) The data represented in figure 1 is interesting, however the effect on transplanted tumor growth is not so impressive, particularly the survival rates are almost similar indicating rather a minor effect of this regimen on transplanted tumor growth and depend on the interpretation of n=5. Moreover, it is unclear if the high-sodium diet (HSD) has other effects on the animals, leading to delayed tumor growth. The authors have to present weights of the animals, food and water intake. Do the different diets have an impact on the general health of the animals? Since it is known that the renin angiotensin system could impact cancer development- could the authors exclude any hormonal effects, related to HSD? Furthermore, and in this line, it is known that HSD could have sex specific effects (Krementsov et al), thus, the authors have to examine the effects in male mice as well. More importantly, allogeneic cancer models used here are of limited value. Does a high salt diet also impact spontaneous tumor models, which mimic human cancer development and the suppressive involvement of MDSCs for instance much better? Is there any effect on angiogenesis? N=5 for key experiments is not sufficient. Dot plot presentation is needed.

2) The authors claim according to figure 2a and supplemental figure 2a, that the high-salt diet induces salt (NaCl) accumulation in tumor and other tissues. However, they only measured osmolality by vapor pressure osmometry. Thus, the authors cannot judge if there is indeed Na⁺ accumulation in these tissues. Other osmolytes like urea or potassium might have accounted for the altered osmolarity (J Clin Invest 2017;127:1944-1959). The authors should use more specific

methods to define the Na⁺ content in those tissues e.g. by Electron microprobe analysis (J Clin Invest. 2013;123:2803-15.)

3) It is a bit surprising that the authors could observe increased levels of IFN γ and TNF α levels of tumor lysates only by ELISA (Fig2e) but not by their protein array (2d). Thus, it is indicated that they should prove this by an independent method (e.g. qPCR) for the depicted parameters. Moreover, it is not defined how they prepared these lysates. In this figure it is further not defined how they defined MDSC and Macrophage and DC population by FACS (2f-1, sf2)) thus representative FACS plots and a gating strategy need to be added.

4) Regarding figure 3, the authors want to make the point that the HSD regimen promotes the differentiation into a more pro-inflammatory state by in vitro culture of ex-vivo isolated MDSC. The authors have to show respective sortings of MDSC after isolation and before in vitro culture (same for PMN-MDSC/sfig 3). What is the rationale to culture these cells for 3 days? The authors should follow the cells over longer period of time and at earlier time points and make a time course. In addition, the array data were taken after 24hrs after cultivation (3c). It be more direct and informative to screen these cells directly after ex vivo isolation. The in vitro culture period could potentially induce severe changes in gene expression. The representative immunofluorescence shown in figure 3f is hard to interpret, since there is now quantification added.

5) In figure 4 the authors claim that HSD enhances immune surveillance by T cells in dependence of MDSC. They claim that the HSD would affect proliferation and function of T cells. However, the authors didn't check for proliferation- they only found a higher frequency of CD4⁺ and CD8⁺ T cells measured in different organs, which could have multiple reasons. In toto the T cell analysis is very superficial. T cell subpopulations should be analyzed more in detail, including activation markers and subpopulations (e.g. Tregs). Moreover, the immunofluorescence for IFN γ is again not quantified and should be backed up by intracellular FACS for infiltrating CD8 & CD4 T cells. Since HSD is known to boost Th17 cells - why they did not measure IL17 and related cytokines? Most, importantly - since the authors speculate that the potential T cell activation might be key for the effect - they have to investigate the HSD effect in CD8 and in addition CD4 T cell deficient models to make a point here.

6) In figure 5 the authors want to make the claim that the observed in vivo effect is due to NFAT5 activation in MDSC. However, again they only show correlative data, based on in vitro assays. To really make a point that NFAT5 is key to their in vivo observation and that MDSC really play a role under these circumstances the authors have to repeat their experiments in MDSC specific conditional NFAT KO mice.

7) Please provide all figure (where applicable) either as dot plot representations or whiskers and dot plots. In general for a lot of experiments the n-number should be increased.

Minor points:

1) The authors should clearly state the number of technical and biological replicates.

2) It is unclear if the depicted data in Figure 1e was quantified?

3) Figure 2f,g,h - the definition of T- and M- MDSC is not clear

4) Figure 3c - the complete array data has to be included

5) Figure 4a - FACS plots should be presented including 'outliers'

6) Figure 5 - it is unclear when and how the MDSC's were isolated

7) Figure 5 h, l, j, k 'none' could be removed, figure 5k: wrong legend for NaCl

8) Manuscript contains many typos and would benefit from proofreading by a native speaker

9) The authors should follow the ARRIVE guideline and report on details for their experimental designs. E.g. Vendor of the chow including catalogue numbers, whether experiments were evaluated in a blinded manner, etc.

Reviewer #2:sodium metabolism and diseases

(Remarks to the Author):

Using an experimental transfer model of melanoma and breast cancer, the authors show in this interesting manuscript that a high salt diet (HSD) reduces tumor growth and improves survival in mice. The authors demonstrate that monocytic- and granulocytic-myeloid derived suppressor cells polarize into anti-tumor macrophages and adopt pro-inflammatory function when exposed to high-salt environmental conditions, which may explain the beneficial salt effect on tumor growth. My concerns include the incomplete characterization of the osmolyte composition of the microenvironment in the tissues and the lack of knowledge of the literature that requires revision. Additional differential insights on the relative contribution of the JNK, nf-kappa b, p38/ERK regulatory cascade and lipopolysaccharide driven cell responses would further increase the significance of this manuscript.

1. TonEBP/Nfat5 responds to salt-driven osmotic stress, not to urea-driven osmotic stress. Osmolality measurements alone do not confirm the presence of osmotic stress, which requires Na(?) -driven hypertonicity. Please provide evidence that the increase in osmolality in thymus, spleen, liver and tumor in HSD-fed animals (Figure 2a) was due to increased sodium storage in the tissues. Na, K, and water content could be analyzed chemically.

As a side, Figure 2A indicate that plasma osmolality was 320 mOsm in LSD- and HSD-treated animals. This finding suggests severe hyperosmolality and/or dehydration of the animals. Please confirm the measurements, and, if correct, provide an explanation why the animals were dehydrated.

2. Macrophage polarization patterns and cell culture experiments (Figure 5): The cell culture experiments suggest that TonEBP might be responsible for anti-tumor polarization patterns in response to high salt. However, the biological signature of the macrophages from tumor tissue also suggests involvement of the JNK-cascade, nf-kappa regulatory processes, and responses to lipopolysaccharides (Figure 5A). The authors should use their in-vitro approaches, including Nfat5 silencing, to clarify the contribution of salt-driven hyperosmolality to these cellular responses mechanistically.

i) Was the JNK-regulatory cascade activated by Nfat-5 expression in response to osmotic stress, or did this tumor-macrophage response occur independent of hypertonicity? Does Nfat-5 silencing in tumor-associated macrophages reduce the JNK-regulatory cascade? Previous reports suggest that salt-driven osmotic stress plays does not promot the JNK-regulatory cascade in an infection model (Cell Metabolism. 2015;21:493-501). Are things different in tumor macrophages?

ii) Was nf-kappaB regulatory cascade avtivated by Nfat-5 expression in response to osmotic stress, or did this tumor-macrophage response occur independent of hypertonicity? Does Nfat-5 silencing in

tumor-associated macrophages reduce the nf kappa b-regulatory cascade? Previous reports suggest that salt-driven osmotic stress plays does not promote the nf kappa b cascade in an infection model (Cell Metabolism. 2015;21:493-501). Are things different in tumor macrophages?

iii) In an infection model (Cell Metabolism. 2015;21:493-501), Nfat5-dependent pro-inflammatory macrophage polarization was driven by promotion of the TLR-4 pathway via p38/ERK kinase. Given the involvement of lipopolysaccharides in the tumor-macrophage response (Figure 5A), do macrophages similarly use the p38/ERK kinase pathway for anti-tumor polarization? Similar to Nfat5/TonEBP silencing, does p38 gene deletion als reduce tnf alpha, il12, and NOS2 levels, and increase Arg1 and Il10 levels in tumor-associated macrophages exposed to sodium-driven hyperosmotic stress?

3. Lines 240-244: The authors erroneously quote references 31 and 32. The relationship between sodium storage and tissue osmolality was not investigated in these studies. The first reports on salt/TonEBP-driven modulation of macrophage function is the description of homeostatic macrophages that regulate tissue sodium content and blood pressure in response to sodium storage in the skin (Nat Med. 2009;15:545-552, Hypertension. 2010;55:755-761, J Clin Invest. 2013; 123: 2803-2815). It is unclear why the authors have decided to neglect these studies. Together with Cell Metabolism. 2015;21:493-501; these would be the correct citations on the relationship between tissue Na storage and hyperosmolality.

Reviewer #3: MDSCs

(Remarks to the Author):

In this manuscript, Authors report that mice fed on a high-salt diet (HSD) had increased concentration of sodium chloride (NaCl) within the tumor environment and the consequent osmotic stress affected tumor growth. Authors describe changes in subsets of myeloid-derived suppressor cells (MDSCs) with a shift towards activated macrophages and PMN and reduction in the immunosuppressive cells. Allegedly, this change also rendered T cells more responsive to stimulation in different districts. Finally, using in vitro experiments Authors showed that silencing NFAT5 reversed the conversion of M-MDSC to inflammatory macrophages (M1 type).

Specific issues

1) Even though the idea is intriguing, I am not sure how and if it can be transferred to the clinical setting. Considering the potential side effects of such diet regimen, it would be admissible only for a limited time; moreover, the advent of checkpoint inhibitors demands for the combination therapy. Is the HSD increasing the effect of checkpoint inhibitors towards established tumors? This obvious question requires to be addressed specifically, also in light of the issue raised on the point #2.

2) The concept that tumor control in HSD is dependent from an adaptive immunity is indirect; there is no experiment in immunocompromised mice or in mice depleted for T lymphocyte subsets. This is key since the antitumor activity might only rely on the macrophage activation.

3) Considering the discrepancies with the ELISA results, the cytokine array relevance is questionable.

4) The protocol for MDSC differentiation is not described in material and method; in particular, it is unclear what factors are used for differentiation (if any) and what the Authors really mean for differentiation. Is cell viability affected after culture?

5) I do not understand the logic behind the selection of markers to evaluate conversion of PMN-MDSCs following HSD. Arg1 is poorly expressed in mouse PMN and ICAM-1 and CXCR4 cannot be considered markers of different stages. A deeper characterization might probably help.

6) In the best case, the NFAT5 silencing experiments only prove that some properties of M-MDSCs can be affected by NaCl in vitro. To prove that the HSD is operating in regulating differentiation of precursors to M-MDSC and macrophages through NFAT5, the only possibility is to use NFAT5 knockout mice. There are different floxed mice already published that can be used for conditional knockout. I suspect that high NaCl might also affect differentiation of normal monocytes through an NFAT5-dependent step.

7) The English must be reviewed throughout the manuscript, some points are difficult to follow.

Response to the reviewers:

To reviewer #1:

Major points:

Q1. The data represented in figure 1 is interesting, however the effect on transplanted tumor growth is not so impressive, particularly the survival rates are almost similar indicating rather a minor effect of this regimen on transplanted tumor growth and depend on the interpretation of n=5. Moreover, it is unclear if the high-sodium diet (HSD) has other effects on the animals, leading to delayed tumor growth. The authors have to present weights of the animals, food and water intake. Do the different diets have an impact on the general health of the animals? Since it is known that the renin angiotensin system could impact cancer development- could the authors exclude any hormonal effects, related to HSD? Furthermore, and in this line, it is known that HSD could have sex specific effects (Krementsov et al), thus, the authors have to examine the effects in male mice as well. More importantly, allogeneic cancer models used here are of limited value. Does a high salt diet also impact spontaneous tumor models, which mimic human cancer development and the suppressive involvement of MDSCs for instance much better? Is there any effect on angiogenesis? N=5 for key experiments is not sufficient. Dot plot presentation is needed.

A1: These are excellent questions. First, we increased the sample size, especially for assessing the survival rates where we put 10 mice per group, to assess the effect of high salt diet (HSD) on tumor growth. We found that HSD markedly inhibited transplanted tumor growth and prolonged the survival rate of tumor-bearing mice (**Fig. 1**). Meanwhile, we examined the weights of tumor-bearing mice, food and water intake during the HSD period. We found that high salt diet significantly increased the food and water intake of the mice (**supplementary Fig. 2a, b**). In

addition, HSD did not significantly affect the body weight of the tumor-bearing mice (**Supplementary Fig. 2c**).

Second, we investigated whether HSD affected the general health of the animals. We found no significant differences in the spleen, liver and kidney indexes between the NSD and HSD group (**Supplementary Fig. 3a-f**). Additionally, HSD applied for 16 days showed no hepatotoxicity and nephrotoxicity, with no abnormality observed in the serum levels of alanine transaminase (ALT), aspartate aminotransferase (AST), blood urea nitrogen (BUN) and creatinine (**Supplementary Fig. 3g-j**).

Third, recent evidence shows that the renin-angiotensin system (RAS) can affect all hallmarks of cancer, and the dysregulation of RAS plays an important role in tumor growth and metastasis (*Pinter and Jain, Science Translational Medicine, 2017, 9(410); Pinter et al, Clinical Cancer Research, 2018, 24(16):3803-3812*). Meanwhile, dietary sodium loading inhibits the systemic RAS, resulting in lower levels of plasma renin and angiotensin II (AngII), among other changes (*Drenjancevic-Peric et al, Kindey & blood pressure research, 2011, 34(1): 1-11*). In the revised study, we investigated the anti-tumor activation of HSD after inhibiting RAS using Angiotensin-converting enzyme inhibitors (captopril). Our data show that: i) indeed, blockade of RAS inhibited tumor growth; but ii) under the inhibition of RAS, HSD also exerted the anti-tumor activity. Both HSD and captopril decrease the level of plasma AngII; however, the less suppression from HSD comparing captopril group was more effective to inhibit the growth of the tumors. Although the data cannot completely exclude the possible involvement of RAS, at least suggesting that the anti-tumor activity of HSD had significant independent anti-tumor effects from RAS (**supplementary Fig. 7a-f**).

Fourth, as suggested, we confirmed that HSD exerted similar anti-tumor efficacy in male and female mice (**Fig. 1**). We also added that HSD did not affect the blood vessels in the tumor (**Fig. 4g, h**).

Fifth, we appreciate the reviewer's suggestion of using spontaneous tumor models. Indeed, relevant studies using such models have indicated a correlation between tumor progression and accumulation of MDSCs in various cancer types

(Kumar et al, *The Journal of clinical investigation*, 2018, 128 (11): 5095-5109; Chun, *Cell reports*, 2015, 12 (2): 244-257; Blattner et al, *cancer research*, 2018, 78 (1): 157-167). Therefore, we speculate that HSD may suppress tumor growth. Nevertheless, having had consulted the animal suppliers as well as experienced researchers on this model, we realized that such an experiment could not be finished within the revision period (even with reasonable extension) and might take over 1 year. It is extremely time-consuming even if only for preparing enough animals. Therefore, we regret to request the reviewer's understanding that it is impossible to do such models for this revision. However, we have discussed the limitations of the allogeneic model in the Discussion part of the revised paper.

Finally, we increased the sample size (n = 10 mice/group) in key experiments (e.g. the anti-tumor activity of HSD, **Fig. 1**; the effect of HSD on pro-inflammation cytokines and MDSCs differentiation, **Fig. 3d-i and Supplementary Fig. 5**; the effect of HSD on T-cell mediated anti-tumor response, **Fig. 5**; and the effect of HSD on the body weight, food and water intake, **supplementary Fig. 2**) and used dot plot to present our results.

Q2. The authors claim according to figure 2a and supplemental figure 2a, that the high-salt diet induces salt (NaCl) accumulation in tumor and other tissues. However, they only measured osmolality by vapor pressure osmometry. Thus, the authors cannot judge if there is indeed Na⁺ accumulation in these tissues. Other osmolytes like urea or potassium might have accounted for the altered osmolality (J Clin Invest 2017;127:1944-1959). The authors should use more specific methods to define the Na⁺ content in those tissues e.g. by Electron microprobe analysis (J Clin Invest. 2013;123:2803-15.)

A2. We thank the reviewer for this important suggestion. To confirm that the increase in osmolality in the thymus, spleen, liver, and tumor in HSD-fed animals was a consequence of the increased sodium storage in these tissues, we used atomic adsorption spectrometry or titration to analyze Na⁺, Cl⁻ and K⁺ content, and found

that HSD promoted Na⁺ and Cl⁻ accumulation in the thymus, spleen and liver, with an especially higher storage in the tumor sites; however, no such difference was observed in the serum (**Fig. 2a, b and supplementary Fig. 4a, b**). Meanwhile, we found no differences in K⁺ and water content in different organs and serum between the NSD and HSD group (**supplementary Fig. 4 c-e**). In addition, we used X-ray fluorescence spectrometer to examine the content of Na⁺ in tumor tissues from both the NSD and HSD group. We found that HSD increased Na⁺ storage in tumor tissues from the HSD group (**Shown in Fig. 2c and supplement Fig. 4f**).

Q3. It is a bit surprising that the authors could observe increased levels of IFN γ and TNF α levels of tumor lysates only by ELISA (Fig2e) but not by their protein array (2d). Thus, it is indicated that they should prove this by an independent method (e.g. qPCR) for the depicted parameters. Moreover, it is not defined how they prepared these lysates. In this figure, it is further not defined how they defined MDSC and Macrophage and DC population by FACS (2f-l, sf2)) thus representative FACS plots and a gating strategy need to be added.

A3. We appreciate the comments. In revising this study, we performed qPCR and found that HSD could up-regulate the levels of IFN- γ and TNF- α in tumor lysates at transcriptional level (**Fig. 3d**). Also in the revised paper, we have described the method of preparing tumor lysates and provided representative FACS plots and a gating strategy for MDSC, macrophages, and DC population.

Q4. Regarding figure 3, the authors want to make the point that the HSD regimen promotes the differentiation into a more pro-inflammatory state by in vitro culture of ex-vivo isolated MDSC. The authors have to show respective sortings of MDSC after isolation and before in vitro culture (same for PMN-MDSC/sfig 3). What is the rationale to culture these cells for 3 days? The authors should follow the cells over longer period of time and at earlier time points and make a time course. In

addition, the array data were taken after 24hrs after cultivation (3c). It be more direct and informative to screen these cells directly after ex vivo isolation. The in vitro culture period could potentially induce severe changes in gene expression. The representative immunofluorescence shown in figure 3f is hard to interpret, since there is now quantification added.

A4. The respective sorting of M-MDSCs and PMN-MDSCs after isolation and before in vitro culture has been added in **supplementary Fig. 15a, b**.

For culture these cells for 3 days, several studies have demonstrated that, to induce MDSCs generation in vitro, cells from bone marrow were cultured in RPMI1640 complete medium (containing 10% FBS) and then stimulated with combinations of GM-CSF and IL-6 (40 ng/ml each) for 4 days (*Zhang et al, Plos one, 2013, 8 (8): e70828*). However, to generate macrophages in vitro, cells from bone marrow were cultured in RPMI1640 complete medium containing 10% FBS and 20 ng/ml GM-CSF for 7 days (*Binger et al, The Journal of clinical investigation, 2015, 125 (11): 4223-4238*). Therefore, we cultured these cells for 3 days to investigate the differentiation of M-MDSC into macrophage. In addition, the differentiation of M-MDSC into macrophages for longer (day 4) and at an earlier point (day 2) was also investigated. As shown in **Fig. 4a, b**, high salt environment promoted the differentiation of M-MDSC into macrophages in a time-dependent manner.

We profiled the mRNA expression in these cells immediately after separation from NSD or HSD mice using Agilent Mouse gene chip analysis without culturing these cells for 24 hours. In the original description, we did not write the details clearly enough and have possibly confused the reviewer.

At last, the quantification of figure 3f has been added to **Fig. 4g, h** in the revised paper.

Q5. In figure 4 the authors claim that HSD enhances immune surveillance by T cells in dependence of MDSC. They claim that the HSD would affect proliferation and function of T cells. However, the authors didn't check for proliferation- they only found a higher frequency of CD4+ and CD8+ T cells measured in different organs, which could have multiple reasons. In toto the T cell analysis is very superficial. T cell subpopulations should be analyzed more in detail, including activation markers and subpopulations (e.g. Tregs). Moreover, the immunofluorescence for IFN γ is again not quantified and should be backed up by intracellular FACS for infiltrating CD8 & CD4 T cells. Since HSD is known to boost Th17 cells - why they did not measure IL17 and related cytokines? Most, importantly - since the authors speculate that the potential T cell activation might be key for the effect - they have to investigate the HSD effect in CD8 and in addition CD4 T cell deficient models to make a point here.

A5. It is an important question. The proportion of CD4⁺ and CD8⁺ T cells within CD45⁺ cells markedly increased in the blood, spleen, and tumor of HSD-treated mice. As suggested, we revised the data using flow cytometry to analyze Ki67 expression after gating CD4⁺ and CD8⁺ T cells isolated from tumor tissues for checking the proliferation of T cells (**Fig. 5e, f**). We further analyzed the subpopulation of the T cells including Treg and Th17 (**Fig. 5g, i**, and **Supplementary Fig. 10a, b**) and quantified the IFN γ with intracellular FACS (**Fig. 5g-j**, and **Supplementary Fig. 10 a**). The expression of TNF- α in Th17 cells was also measured (**Supplementary Fig. 10 c**).

According to the reviewer's wise suggestion on the animal model, we grafted two tumor models (4T1 and B16F10 tumor) to BALB/c null mice to examine whether the anti-tumor effects of HSD is dependent on T cell activation. The results were shown in **supplementary Fig. 10d-g**.

Q6. In figure 5 the authors want to make the claim that the observed in vivo effect

is due to NFAT5 activation in MDSC. However, again they only show correlative data, based on *in vitro* assays. To really make a point that NFAT5 is key to their *in vivo* observation and that MDSC really play a role under these circumstances the authors have to repeat their experiments in MDSC specific conditional NFAT KO mice.

A6. We appreciate this suggestion. The ideal way to examine it is to use MDSC-specific conditional NFAT5 knockout mice. Unfortunately, we understood from the Jackson Laboratory, which is the only possible supplier of this KO model, that such mice were still unavailable. Thus, it is technically impossible for us to obtain NFAT5 KO mice. However, we adopted an alternative way to validate the key role of NFAT5 in mediating the effect of HSD on M-MDSC-macrophage differentiation. The tumor-bearing mice were intravenously administered with lentiviruses of siRNA-NFAT5-LV (lentivirus carrying NFAT5 siRNA under Ly-6C promoter) every two days for 8 injections to knock down the NFAT5 *in vivo*. The proportion of M-MDSCs and macrophage as well as M-MDSCs function in tumor tissues were then measured (**Supplementary Fig 13 a-d**). Meanwhile, the M1 marker (TNF- α , IL-12, and NOS2) and M2 marker (IL-10 and Arg1) were also measured under the situation of NFAT5 deficiency (**Supplementary Fig. 13e, f**). All the results obtained from this group of experiments agree well with our previous observation that NFAT5 mediated the above response in the mice treated with HSD.

Q7. Please provide all figure (where applicable) either as dot plot representations or whiskers and dot plots. In general for a lot of experiments the n-number should be increased.

A7. In the revised manuscript, we have provided the figures in the form suggested by the reviewer. We also increased the n-numbers in most of the experiments to 10 in the revised paper.

Minor points:

Q1. The authors should clearly state the number of technical and biological replicates.

A1. We have stated such information in the revised paper.

Q2. It is unclear if the depicted data in Figure 1e was quantified?

A2. Thanks for the reminder. It was done in the revised paper.

Q3. Figure 2f,g,h - the definition of T- and M- MDSC is not clear

A3. We thank the reviewer for this reminder. We gated CD11b⁺Gr-1⁺ cells as T-MDSCs and CD11b⁺Ly6-C⁺ cells as M-MDSCs (**Fig. 3**). We have added it in the revised paper.

Q4. Figure 3c - the complete array data has to be included

A4. We have included the complete array data (**Fig. 4c, d**) in the revised paper.

Q5. Figure 4a - FACS plots should be presented including 'outliers'

A5. We have presented the plots including 'outliers' in the revised paper.

Q6. Figure 5 - it is unclear when and how the MDSC's were isolated

A6. We have added such information in the Methods part of the revised manuscript.

Q7. Figure 5 h, i, j, k 'none' could be removed, figure 5k: wrong legend for

NaCl

A7. We have corrected these mistakes.

Q8. Manuscript contains many typos and would benefit from proofreading by a native speaker

A8. We thank the reviewer for this advice. To improve the English writing, we asked the Nature Research Editing Service, which was recommended by the editor to proofread our manuscript. We uploaded the certification from their office as supplementary supporting material in the system.

Q9. The authors should follow the ARRIVE guideline and report on details for their experimental designs. E.g. Vendor of the chow including catalogue numbers, whether experiments were evaluated in a blinded manner, etc.

A9. We thank the reviewer for this reminder and have reported all these details in the revised manuscript.

To reviewer #2:

Q1. TonEBP/Nfat5 responds to salt-driven osmotic stress, not to urea-driven osmotic stress. Osmolality measurements alone do not confirm the presence of osmotic stress, which requires Na(?) -driven hypertonicity. Please provide evidence that the increase in osmolality in thymus, spleen, liver and tumor in HSD-fed animals (Figure 2a) was due to increased sodium storage in the tissues. Na, K, and water content could be analyzed chemically. As a side, Figure 2A indicate that plasma osmolality was 320 mOsm in LSD- and HSD-treated animals. This finding suggests severe hyperosmolality and/or dehydration of the animals. Please confirm the

measurements, and, if correct, provide an explanation why the animals were dehydrated.

A1. This is a very interesting point. First, we used atomic adsorption spectrometry or titration to analyze Na^+ , Cl^- and K^+ content, and found that HSD promoted Na^+ and Cl^- accumulation in the thymus, spleen, and liver, with an especially higher storage in the tumor site; but we observed no differences in the serum (**Fig. 2a, b and supplementary Fig. 4a, b**). We found no differences in the level of K^+ and water content in the thymus, spleen, liver, and tumor between the NSD and HSD group (**supplementary Fig. 4 c-e**). Also, by using X-ray fluorescence spectrometer to examine the concentration of Na^+ , we found that HSD enhanced Na^+ storage in the tumor of the HSD group (**Shown in Fig. 2c and supplement Fig. 4f**).

In addition, we checked our experimental records and noticed that the serum osmolality was 300~310 mOsm in LSD- and HSD-treated animals – both are below 320 mOsm and in the normal range of osmotic pressure (*Go et al, Proceedings of the National Academy of Sciences of that United States of America, 2004, 101 (29): 10673-10678, Aoka et al, Cardiovascular Research, 2014, 104 (2): 326-336*). When re-measuring the osmotic pressure of different organs of the tumor-bearing mice, we found that the plasma osmolality from LSD and HSD group was also 300~310 mOsm (shown in **Fig. 2d and supplementary Fig. 4g**). Meanwhile, no significant pathological changes and mortality were observed during all our original and new experiments. We believe that the animals were not in a dehydrate state.

2. Macrophage polarization patterns and cell culture experiments (Figure 5):
The cell culture experiments suggest that TonEBP might be responsible for anti-tumor polarization patterns in response to high salt. However, the biological signature of the macrophages from tumor tissue also suggests

involvement of the JNK-cascade, nf-kappa regulatory processes, and responses to lipopolysaccharides (Figure 5A). The authors should use their in-vitro approaches, including Nfat5 silencing, to clarify the contribution of salt-driven hyperosmolality to these cellular responses mechanistically.

i) Was the JNK-regulatory cascade activated by Nfat-5 expression in response to osmotic stress, or did this tumor-macrophage response occur independent of hypertonicity? Does Nfat-5 silencing in tumor-associated macrophages reduce the JNK-regulatory cascade? Previous reports suggest that salt-driven osmotic stress plays does not promot the JNK-regulatory cascade in an infection model (Cell Metabolism. 2015;21:493-501). Are things different in tumor macrophages?

ii) Was nf-kappaB regulatory cascade avtivated by Nfat-5 expression in response to osmotic stress, or did this tumor-macrophage response occur independent of hypertonicity? Does Nfat-5 silencing in tumor-associated macrophages reduce the nf kappa b-regulatory cascade? Previous reports suggest that salt-driven osmotic stress plays does not promote the nf kappa b cascade in an infection model (Cell Metabolism. 2015;21:493-501). Are things different in tumor macrophages?

iii) In an infection model (Cell Metabolism. 2015;21:493-501), Nfat5-dependent pro-inflammatory macrophage polarization was driven by promotion of the TLR-4 pathway via p38/ERK kinase. Given the involvement of lipopolysaccharides in the tumor-macrophage response (Figure 5A), do macrophages similarly use the p38/ERK kinase pathway for anti-tumor polarization? Similar to Nfat5/TonEBP silencing, does p38 gene deletion als reduce tnf alpha, il12, and NOS2 levels, and increase Arg1 and Il10 levels in tumor-associated macrophages exposed to sodium-driven hyperosmotic

stress?

A2. We thank the reviewer for these serial questions and have accordingly taken the advice to perform new experiments during the revision. We have carefully examined the signaling pathways involved in MDSCs responding to HSD stimulation. Our new data highlight that HSD promotes M-MDSCs differentiation and function transformation through p38/MAPK-dependent NFAT5 activation. First, the activation by an additional 40 mM NaCl for 1 hour could not affect JNK phosphorylation in M-MDSCs. However, a 24-h HSD stimulation upregulates JNK phosphorylation in M-MDSCs, while NFAT5 silencing blunted JNK phosphorylation – indicating that the conditions of HSD indirectly mediates JNK signaling (shown in **Fig. 7o** and **supplementary Fig. 11a**). Second, a high-salt stimulation of 1 h could not affect, but that of 24 hours upregulated, NF-κB activity in M-MDSCs; however, knocking down of NFAT5 down-regulated NF-κB activity, indicating that high salt conditions indirectly mediates NF-κB activity (**Fig. 7o** and **supplementary Fig. 11b**). Third, a high-salt stimulation of 1 h augments the phosphorylation of p38/MAPK (shown in **Fig. 7b**), and high-salt conditions enhances NFAT5 expression (shown in **Fig. 7i**), but knock down of p38 could inhibit high salt-induced NFAT5 expression (shown in **Fig. 7j**). In addition, similar to NFAT5 silencing, p38 silencing could also blunt high salt-induced M-MDSC differentiation and function (shown in **Supplementary Fig. 12**).

Taking the above data together, we conclude that – i) HSD enhances MDSCs differentiation and function via p38/MAPK and NFAT5 expression, and ii) the JNK and NF-κB-regulatory cascades are indirectly activated by p38/MAPK and NFAT5 expression in response to osmotic stress.

3. Lines 240-244: The authors erroneously quote references 31 and 32. The relationship between sodium storage and tissue osmolality was not

investigated in these studies. The first reports on salt/TonEBP-driven modulation of macrophage function is the description of homeostatic macrophages that regulate tissue sodium content and blood pressure in response to sodium storage in the skin (Nat Med. 2009;15:545-552, Hypertension. 2010;55:755-761, J Clin Invest. 2013; 123: 2803-2815). It is unclear why the authors have decided to neglect these studies. Together with Cell Metabolism. 2015;21:493-501; these would be the correct citations on the relationship between tissue Na storage and hyperosmolality.

A3. We thank the reviewer for this reminder and have updated citations in the revised paper.

To reviewer #3:

Q1. Even though the idea is intriguing, I am not sure how and if it can be transferred to the clinical setting. Considering the potential side effects of such diet regimen, it would be admissible only for a limited time; moreover, the advent of checkpoint inhibitors demands for the combination therapy. Is the HSD increasing the effect of checkpoint inhibitors towards established tumors? This obvious question requires to be addressed specifically, also in light of the issue raised on the point #2.

A1. We thank the reviewer for this excellent comment. MDSCs promote tumor immune escape by inhibiting T cell-mediated anti-tumor response (*Chiu et al, Nature communications, 2017, 8 (1): 517*), thereby reducing the efficacy of the T cell-targeting immunotherapies. PD-1 is among the most extensively studied immune checkpoint receptors for cancer immunotherapy (*Pardoll DM, Nature reviews cancer, 2012, 12 (4): 252-264*). Anti-PD-1 antibody (Nivolumab), approved by US FDA for the treatment of melanoma,

has shown notable long-term benefits in cancer patients (*Pardoll DM, Nature reviews cancer, 2012, 12 (4): 252-264*). Therefore, we hypothesized that HSD would impair MDSCs function, enhance T-cell penetration into tumors, and eventually improve the efficacy of such therapeutic strategies.

According to the reviewer's valuable suggestions, we have performed new experiments using HSD in combination with anti-PD-1 antibody in the two tumor models. As shown in **Fig. 6a-d**, the combination of HSD and the anti-PD-1 antibody achieved the highest tumor suppressive response in 4T1 and B16F10 tumor models and significantly increased the infiltration of T cells (shown in **Fig. 6 e-g**). These findings suggest that HSD can promote the efficacy of PD-1-based therapy.

However, despite the promising results from our study, the clinical translation of this approach will rely on further tests that should involve various clinically-relevant models and thorough assessment of in vivo safety. The length, intensity, and frequency of HSD should also be fine-tuned under careful observation.

Q2. The concept that tumor control in HSD is dependent from an adaptive immunity is indirect; there is no experiment in immunocompromised mice or in mice depleted for T lymphocyte subsets. This is key since the antitumor activity might only rely on the macrophage activation.

A2. We agree with the reviewer on this point. We have established two tumor models in BABL/C null mice using 4T1 and B16F10 cells. As shown in **supplementary Fig. 10d-g**, there was no significant difference in the tumor growth index between the NSD and HSD group, suggesting that tumor control in HSD was dependent of adaptive immunity.

Q3. Considering the discrepancies with the ELISA results, the cytokine array

relevance is questionable.

A3. As per suggestion, we have examined the levels of IFN- γ and TNF- α in tumor tissues by using qPCR assay. We found that HSD could up-regulate IFN- γ and TNF- α expression in tumor tissues (shown in **Fig. 3d**) – consistent with the ELISA data.

Q4. The protocol for MDSC differentiation is not described in material and method; in particular, it is unclear what factors are used for differentiation (if any) and what the Authors really mean for differentiation. Is cell viability affected after culture?

A4. These are excellent questions. First, the protocol for MDSC differentiation is described in detail in the method of the revised paper. Second, GM-CSF (10 ng/ml) was used for differentiation of M-MDSCs isolated from tumor and cultured in RPMI-1640 medium containing 10% FBS. Third, as shown in the revised **Fig. 4a, b and Supplementary Fig. 8a, b**, we consider the expression of marker F4/80 and CD11c as the indicators of cell differentiation [*Kumar et al, Immunity, 2016, 44 (2): 303-315*]. The F4/80⁺CD11b⁺ cells are macrophages and the CD11c⁺CD11b⁺ cells are DC. Finally, we measured cell viability in this process; we found that the viability of these cells was greater than 90% and not significantly influenced (shown in **supplement Fig. 8c, d**).

Q5. I do not understand the logic behind the selection of markers to evaluate conversion of PMN-MDSCs following HSD. Arg1 is poorly expressed in mouse PMN and ICAM-1 and CXCR4 cannot be considered markers of different stages. A deeper characterization might probably help.

A5. First, when we selected markers, we referred to the recent literature published in highly recognized journals (*Veglia et al, Nature immunology,*

2018, 19 (2): 108-119; Fridlender et al, Cancer cell, 2009, 16 (3): 183-194; Yang et al, Journal of molecular cell biology, 2013, 5 (3): 207-209). Second, to further address this issue, we evaluated a wider range of markers including TNF- α , ICAM-1, PGE2, Arg1, CCL2, and CCL5 (shown in **supplementary Fig. 9a-d**).

Q6. In the best case, the NFAT5 silencing experiments only prove that some properties of M-MDSCs can be affected by NaCl in vitro. To prove that the HSD is operating in regulating differentiation of precursors to M-MDSC and macrophages through NFAT5, the only possibility is to use NFAT5 knockout mice. There are different floxed mice already published that can be used for conditional knockout. I suspect that high NaCl might also affect differentiation of normal monocytes through an NFAT5-dependent step.

A6. We highly appreciate the advice and have used an alternative approach to address this question.

First, we agree that the ideal way to do so is to use MDSCs-specific conditional NFAT5 knockout mice. Unfortunately, we cannot obtain NFAT5 KO mice. The only supplier is the Jackson Laboratory in the USA. However, the mice had been unavailable throughout this period according to their website. It is also unrealistic to newly establish a NFAT5 knockout mouse within the period of revision.

Second, our data indicated that HSD could promote M-MDSCs differentiation into macrophages through up-regulating NFAT5 expression. Therefore, we used an alternative way to further verify that NFAT5 was key in response to HSD to regulating the differentiation of M-MDSC to macrophages. The tumor-bearing mice were intravenously administered with lentiviruses of siRNA-NFAT5-LV (lentivirus carrying NFAT5 siRNA under Ly-6C promoter) every two days for 8 injections to knock down the NFAT5 in vivo. The proportion of M-MDSCs and macrophages, as well as the M-MDSCs function, in the tumor tissue was then measured (**Supplementary Fig 13**

a-d). Meanwhile, the M1 (TNF- α , IL-12, and NOS2) and M2 markers (IL-10 and Arg1) were also measured under the situation of NFAT5 deficiency (**Supplementary Fig. 13e, f**). All the results obtained from this experiment were consistent with our previous observation that NFAT5 mediated the responses of the mice to HSD.

Third, to address whether HSD also affects the differentiation of normal monocytes through an NFAT5-dependent mechanism, we isolated normal monocytes from the blood from healthy mice and cultured them in RPMI-1640 medium containing 10% FBS. After knocking down NFAT5, we stimulated these cells with GM-CSF and an additional 40 mM NaCl for 3 days. The generation of Macrophages were measured by flow cytometry. As shown in **Supplementary Figure 14**, HSD also enhanced monocytes differentiation into macrophages, and down-regulation of NFAT5 could inhibit the generation of high NaCl-induced macrophages, suggesting that HSD could promote the differentiation of normal monocytes into macrophages through an NFAT5-dependent way.

Q7. The English must be reviewed throughout the manuscript, some points are difficult to follow.

A7. We thank the reviewer for this reminder. To improve the English writing, we asked the Nature Research Editing Service, which was recommended by the editor to proofread our manuscript. We uploaded the certification from their office as supplementary supporting material in the system.

Reviewers' comments:

Reviewer #1 (Remarks to the Author):

The authors have performed a lot of experiments and the manuscript has improved substantially. However, several discrepancies which shed doubts on the robustness of the data came up. Other issues were not addressed. Some of them are of major importance!

1. I asked the authors to provide ALL figures as dot plots. Indeed, several, but not all figures were changed. Unfortunately, I'm really worried about the data in Fig. 5e & f. Fig. 5e (HSD/CD8 shows 45.7%). However, this value does not occur in the quantification in 5f. Therefore, I would like to ask you for the original data in Fig. 5f.

2. Further doubts on the robustness of the data is derived from the discrepancy between the last version (stating that they worked with cultured material) and the current version (stating that they worked with freshly isolated material). In the last version the array data (Fig. 3c old says "(c) the purified tumor-infiltration M-MDSC from NSD or HSD group were cultured in 821 RPMI-1640 medium containing 10% FBS for 24h. mRNA 822 expression was examined using Agilent Mouse GE v2 Microarrays (4 \times 44k format). 823 mRNA expression levels of representative HSD-regulated genes involved in function 824 in MDSCs from HSD group. n = 5 mice per group.") and now the current version suddenly says "(c, d) mRNA expression was examined using Agilent Mouse GE v2 Microarrays (4 \times 44k format) after tumour-infiltrating M-MDSCs were isolated from the NSD or HSD group. DEGs in the chip analysis and mRNA expression levels of representative HSD-regulated genes involved in MDSC function from the HSD group; n = 5 or 6 mice per group."

3. Another suspicious point is the discrepancy in fig. 3f-old and fig. 4g-new. The pictures are suddenly in an opposite order.

4. Further, why did fig. 4e old disappear from the new version without comment?

5. Further, I asked the authors to follow the ARRIVE guidelines. I appreciate that the authors added some more information, but their statement "A9. We thank the reviewer for this reminder and have reported all these details in the revised manuscript." is not justified.

6. To understand immune phenotypes in animal studies, it is essential to get access to the exact information (catalogue number, vendor) for the chow used, since conventional diets and purified diet affect the immune host axis. The authors failed to report on this. Please provide this information for all animal series.

7. Along this line, they do not provide any information, which chow was used for the Balb/C null experiments and whether these immune compromised mice needed different housing and feeding conditions. Further, information about Balb/C null mice are completely missing. What are Balb/C null mice - Rag1 null mice? It is not clear to me what these mice are. Therefore, it is impossible to evaluate this set of data.

8. I asked in the previous revision to increase the n number, which was done for the majority of experiments. However, two major sets of experiments (Balb/C null and PD-1) are again underpowered. Especially, the Balb/C null experiment (Suppl. S10e) might be misinterpreted due to the low n number. Furthermore, the authors need to show the time course of tumour size development for the PD-1 and Balb/C studies.

9. Statistical concerns: the authors do not report whether they tested all data sets for normal distribution. Looking at the data, some sets look like not being normally distributed. Then the statistical test used would be not correct. This needs to be done. Also, the precise description about replicates is need to ALL experiments. No information is given on blinding during quantification of photos and flow cytometry data.

10. The new FACS data on T cell subpopulations seem unusual (Fig. 5g-l, S10a-c). The representative data look unusual and gates set seem to overestimate % of positive populations e.g. IFN γ and particularly for FoxP3 and IL-17. Did the authors gate out dead cells? If not, this may lead to false-positive results for intracellular FACS and could explain the unusual high values. The authors need to present the complete gating strategy and controls with the proper isotype controls. This also applies for S6.

11. The hypertonicity data is interesting. However, expressing e.g. Na $^{+}$ / dry weight reflects sodium content. In the abstract, the authors report on increased local sodium concentration. This can be done by showing Na $^{+}$ / wet weight. This should be provided for all ions in the main figure and supplement.

12. Page 7: Description for M-MDSC is mixed up with PMN-MDSC. Please correct this. Same applies for fig. 3f & 3g.

13. Immunostains for CD31 in fig. 4g are not visible. Please provide more convincing photos.

14. Please shift fig. 3a to the supplement and enlarge it and provide a clear labeling of the respective dots.

15. Why don't the authors report on tumour weight and size in the NFAT5 silencing study? Further, a proof-of-concept experiment, which can be performed if NFAT5 KO mice are not available, is the MDSC depletion with anti-Gr1 antibodies. This experiment would strengthen the whole concept.

Minor:

- S2a needs to be food intake not feed intake.
- Some axis need to be subdivided (broken axis) that low and high values can be seen.

Reviewer #2 (Remarks to the Author):

The authors have satisfactorily answered all my comments.

Reviewer #3 (Remarks to the Author):

No more comments

Reviewers' comments:

Reviewer #1 (Remarks to the Author):

Q1: I asked the authors to provide ALL figures as dot plots. Indeed, several, but not all figures were changed. Unfortunately, I'm really worried about the data in Fig. 5e & f. Fig. 5e (HSD/CD8 shows 45.7%). However, this value does not occur in the quantification in 5f. Therefore, I would like to ask you for the original data in Fig. 5f.

A1: We thank the reviewer for this timely reminder. All figures as dot plots have been provided in the revised paper.

We apologize for this mistake. The expression of Ki67 in representative flow cytometry analysis in Fig. 5e old were marked incorrectly due to our negligence, which should mark 39.8 not 45.7. Therefore, about Fig. 5e&f, especially HSD/CD8, we repeated these experiment in the past three months, found that the proliferation status of CD4⁺ and CD8⁺ T cells within tumours was greater in HSD-treated animals than in NSD-treated animals, and the new data about the proliferation status of CD8⁺ T cells within tumours in the 4T1 tumour model were shown in the **Figure 5e, f** in the revised paper. Meanwhile, the original data about the proliferation of CD8⁺ T cells in tumour sites in 4T1 tumour model in Figure 5f have been provided in supplementary materials.

Q2: Further doubts on the robustness of the data is derived from the discrepancy between the last version (stating that they worked with cultured material) and the current version (stating that they worked with freshly isolated material). In the last version the array data (Fig. 3c old says "(c) the purified tumor-infiltration M-MDSC from NSD or HSD group were cultured in 821

RPMI-1640 medium containing 10% FBS for 24h. mRNA 822 expression was examined using Agilent Mouse GE v2 Microarrays (4Å~44k format). 823 mRNA expression levels of representative HSD-regulated genes involved in function 824 in MDSCs from HSD group. n = 5 mice per group.”) and now the current version suddenly says “(c, d) mRNA expression was examined using Agilent Mouse GE v2 Microarrays (4×44k format) after tumour-infiltrating M-MDSCs were isolated from the NSD or HSD group. DEGs in the chip analysis and mRNA expression levels of representative HSD-regulated genes involved in MDSC function from the HSD group; n = 5 or 6 mice per group.”

A2: We apologize for confusing expressions. In the last review, this problem has been mentioned. Actually, we profiled the mRNA expression in these cells immediately after separation from NSD or HSD mice using Agilent Mouse gene chip analysis without culturing these cells for 24 hours. In the original description, we did not write the details clearly enough and have possibly confused the reviewer.

Q3: Another suspicious point is the discrepancy in fig. 3f-old and fig. 4g-new. The pictures are suddenly in an opposite order.

A3: We apologize for this mistake. In this study, we report that high-salt intake inhibits tumour growth in mice by activating antitumour immune surveillance through modulating the activities of MDSCs. HSD could promoted M-MDSCs differentiated into antitumour macrophages and PMN-MDSCs adopted pro-inflammatory functions, thereby reactivating the antitumour actions of T cells. In figure 4a-f, HSD could promote M-MDSCs differentiated into macrophages, and display anti-tumour phenotype by increased IL-12, TNF- α , NOS2 expression and decreased IL-10 and Arg1 expression in the tumour tissues of HSD-fed mice. In the section of the result, we described that “Immunofluorescent staining also demonstrated significantly increased IL-12 expression and decreased IL-10 expression in the tumour tissues of

HSD-fed mice. Moreover, the number of F4/80-positive macrophages in tumour tissues from the HSD group was remarkably higher than that in tumour tissues from the NSD group (Fig. 4g, h).” Unfortunately, the images of immunofluorescent staining reflected the results in the Figure 3f-old were placed in an opposite order due to our negligence. We found this mistake during the first revision, and in Figure 4e-new, we corrected them in the revised paper.

Q4: Further, why did fig. 4e old disappear from the new version without comment?

A4: We apologize for that Figure 4e old disappear from the new version without comment. Although the immunofluorescence of IFN- γ in tumour tissues in figure 4e old could indicate that a high activated state of these intratumoral T cells in high salt-treated tumour-bearing mouse, they did not quantify the amount of IFN- γ secreted by infiltrating CD8 & CD4 T cells in tumour sites. In the last review, the reviewers recommended that the content of IFN- γ in tumour tissues should be backed up by intracellular FACS for infiltrating CD8 & CD4 T cells. Therefore, we examined the frequency of IFN- γ -producing CD4⁺ and CD8⁺ T cells in the spleen and tumour by flow cytometry (the data shown in **Figure 5g-i**), and the Figure 4e old were deleted in the revised paper.

In addition, the experiment about the frequencies of CD4⁺ T and CD8⁺ T cells within CD45⁺ cells in spleen of 4T1 tumour model was repeated, and the new data were displayed in **Figure 5a, c** in the revised paper, because the representative flow cytometry analysis (HSD group) in spleen of 4T1 tumour model were misplaced in Fig. 5a-old.

Q5: Further, I asked the authors to follow the ARRIVE guidelines. I appreciate that the authors added some more information, but their statement “A9. We thank the reviewer for this reminder and have reported all these details in the

revised manuscript.” is not justified.

A5: We apologize for this mistake. Animal experiments were reported in compliance with the ARRIVE guidelines (McGrath *et al.*, 2015), and more information have been added in the revised paper. At last, the ARRIVE guidelines checklist have been provided in the revised supplementary materials.

Q6: To understand immune phenotypes in animal studies, it is essential to get access to the exact information (catalogue number, vendor) for the chow used, since conventional diets and purified diet affect the immune host axis. The authors failed to report on this. Please provide this information for all animal series.

A6: We thank the reviewer for this timely reminder. Female and male C57BL/6J and BALB/C mice (6-8 weeks old) were fed chow diet (Rodent diet, 1010041, Shoobree, Xietong Organism, Jiangsu, China), and the Balb/c-*nu/nu* mice were fed chow diet (Rodent diet, 1010019, Shoobree, Xietong Organism, Jiangsu, China). All mouse are free access to pellet food and water in plastic cages, and maintained in specific pathogen-free (SPF) conditions in the animal laboratory at Nanjing University at 22-24 °C with a 12 h light-dark cycle, which has been added into the revised paper.

Q7: Along this line, they do not provide any information, which chow was used for the Balb/C null experiments and whether these immune compromised mice needed different housing and feeding conditions. Further, information about Balb/C null mice are completely missing. What are Balb/C null mice - Rag1 null mice? It is not clear to me what these mice are. Therefore, it is impossible to evaluate this set of data.

A7: We thank the reviewer for this timely reminder. In the Balb/c null experiments, the Balb/c-*nu/nu* mice were fed chow diet (Rodent diet, 1010019, Shoobree, Xietong

Organism, Jiangsu, China), free access to pellet food and water in plastic cages, and maintained in specific pathogen-free (SPF) conditions in the animal laboratory at Nanjing University at 22-24 °C with a 12 h light-dark cycle, which has been added into the revised paper.

The Balb/c null mice is the Balb/c-*nu/nu* mice (Athymic female nude mice), not Rag1 null mice, which has been added into the revised paper.

8. I asked in the previous revision to increase the n number, which was done for the majority of experiments. However, two major sets of experiments (Balb/C null and PD-1) are again underpowered. Especially, the Balb/C null experiment (Suppl. S10e) might be misinterpreted due to the low n number. Furthermore, the authors need to show the time course of tumour size development for the PD-1 and Balb/C studies.

A8: We thank the reviewer for this timely reminder. We increased the sample size on two major sets of experiments (Balb/C null and PD-1) where we put 10 mice per group to assess the effect of high salt diet (HSD) on tumor growth, and found that there was no significant difference in the tumor growth index between the NSD and HSD group in two tumor models in BALB/C null mice using 4T1 and B16F10 cells, suggesting that tumor control in HSD was dependent of adaptive immunity (**Supplementary Fig. 12a-d**). In addition, the combination of HSD and the anti-PD-1 antibody achieved the highest tumor suppressive response in 4T1 and B16F10 tumor models and significantly increased the infiltration of T cells (shown in **Fig. 6 a-e**), suggesting that HSD can promote the efficacy of PD-1-based therapy.

At last, the results of the time course of tumour size development for the PD-1 and Balb/C studies have been provided in the revised **Fig. 6c and Supplementart Fig. 12d, f**.

Q9: Statistical concerns: the authors do not report whether they tested all data sets for normal distribution. Looking at the data, some sets look like not being normally distributed. Then the statistical test used would be not correct. This needs to be done. Also, the precise description about replicates is need to ALL experiments. No information is given on blinding during quantification of photos and flow cytometry data.

A9: This is a very important reminder. About the statistical concerns, we examined all data by IBM SPSS Statistics software, and found some data did not be normally distributed . Therefore,we performed several new experiments about these results, and the new data are shown in **Supplementart Fig. 3 and Supplementart Fig. 8a-c** in the revised paper.

The precise description about replicates in all experiments has been added in the figure legend in the revised paper. Meanwhile, the information on blinding during quantification of photos and flow cytometry data have be described them more clearly in the methods of the revised paper.

Q10. The new FACS data on T cell subpopulations seem unusual (Fig. 5g-I, S10a-c). The representative data look unusual and gates set seem to overestimate % of positive populations e.g. IFN γ and particularly for FoxP3 and IL-17. Did the authors gate out dead cells? If not, this may lead to false-positive results for intracellular FACS and could explain the unusual high values. The authors need to present the complete gating strategy and controls with the proper isotype controls. This also applies for S6.

A10: We thank the reviewer for this excellent suggestion. We did not gate out the dead cells in the FACS data on T cell subpopulations and MDSCs in the last revision. According to your opinion, we firstly gated out the dead cells by 7-AAD or Fixability Dye eFluor 520 (FVD eFluor 520), and test MDSCs, T cells subpopulations and its

function by flow cytometry. The new FACS data on MDSCs, T cell subpopulations and its functions was shown in **Fig. 3c-g, and Fig. 5g-j**. Meanwhile, the complete gating strategy and controls with the proper isotype controls were shown in **Supplementary Fig. 7 and Supplementary Fig. 11** in the revised paper.

Q11. The hypertonicity data is interesting. However, expressing e.g. Na⁺/ dry weight reflects sodium content. In the abstract, the authors report on increased local sodium concentration. This can be done by showing Na⁺/ wet weight. This should be provided for all ions in the main figure and supplement.

A11: We thank the reviewer for the excellent suggestion. All ions concentration (Na⁺/wet weight, K⁺/wet weight and Cl⁻/wet weight) have been provided in the revised paper.

Q12. Page 7: Description for M-MDSC is mixed up with PMN-MDSC. Please correct this. Same applies for fig. 3f & 3g.

A12: We apologize for this mistake. We have now corrected them in the revised paper.

Q13. Immunostains for CD31 in fig. 4g are not visible. Please provide more convincing photos.

A13: We thank the reviewer for this timely reminder, the more convincing immunostains for CD31 have been provided in the revised Figure 4g.

Q14. Please shift fig. 3a to the supplement and enlarge it and provide a clear labeling of the respective dots.

A14: We thank the reviewer for this timely reminder, the Figure 3a has been shifted to the **supplementary Fig. 5**, and a clear labeling of the respective dots have been provided in the **supplementary Table 4**.

Q15. Why don't the authors report on tumour weight and size in the NFAT5 silencing study? Further, a proof-of-concept experiment, which can be performed if NFAT5 KO mice are not available, is the MDSC depletion with anti-Gr1 antibodies. This experiment would strengthen the whole concept.

A15: We thank the reviewer for this timely reminder. When we completed this experiment, the tumour weight and size in the NFAT5 silencing study has been recorded and imaged. We apologize for the results were not reported in the last revised paper. Now, the tumour weight and size in the NFAT5 silencing study have been added into the new revised paper (shown in the revised **Supplementary Fig. 15b-d**).

We thank the reviewer for this excellent suggestion. In the MDSC depletion with anti-Gr-1 antibodies model, mice bearing implanted tumour were injected i.p. with anti-Gr-1 monoclonal antibody or isotype control to eliminate MDSCs every two days during NSD or HSD treatment. We found that depletion of MDSC in tumour-bearing mice by anti-Gr-1 monoclonal antibody could markedly abolish the anti-tumour activity of HSD (shown in the revised **Supplementary Fig.16**), which strengthens the whole concept that high-salt intake inhibits tumour growth in mice by activating antitumour immune surveillance through modulating the activities of MDSCs.

Minor:

Q1: S2a needs to be food intake not feed intake.

A1: We thank the reviewer for this timely reminder. We have now corrected them in the revised paper.

Q2: Some axis need to be subdivided (broken axis) that low and high values can be seen.

A2: We thank the reviewer for this timely reminder, and some axis need to be subdivided (broken axis) has been made in the revised paper.

Reviewers' comments:

Reviewer #1 (Remarks to the Author):

As you realized from my previous review, I was concerned about the robustness of your data. In the current version you could not rise my confidence. It is a weak procedure always to mention "we apologize for the mistake" or "due to our negligence".

It is your duty to provide robust convincing data. I was shocked after you provided a dot plot for figure 3A and B to see that there are only two technical replicates, which were statistically tested. This is unacceptable and cannot be tested by t-test.

Another example was the answer to Q1. How can it happen that a value of 45.7 is suddenly 39.8? How can it be that the previous quantification was not representing this value? Again, the excuse "due to our negligence" is not a good scientific procedure. Moreover, the new quantification does not visualize a value of 39.8? You mention that you repeated experiments. It seems that you still have the same n-number. The material used is not really suitable for repeating experiments. In the best case, a re-analysis is doable. It is not the duty of a reviewer to be a detective to find out what was done. You NEED to report details that one can understand it.

In fig. 3d-e, the dotted values differ between the last and the current submission. Again, there is no explanation given why this is the case. This habit is from my perspective unacceptable.

After these few examples, I stopped reviewing further. Precision in reporting is a high value for science. Without it, science is doubtful.

Response to the Reviewer

Reviewer #1 (Remarks to the Author):

Q1. It is your duty to provide robust convincing data. I was shocked after you provided a dot plot for figure 3A and B to see that there are only two technical replicates, which were statistically tested. This is unacceptable and cannot be tested by t-test.

A1: We completely agree with the reviewer and are sorry for any confusion caused. Here, we clarify this issue by explaining the details of these data and showing the original data images:

- 1) In this experiment, we measured cytokine levels in different tissue using a Proteome Profiler Mouse XL Cytokine Array (ARY028, R&D Systems, Inc.), which is a powerful and efficient tool for such analysis, has been commonly used in studies published in recognised journals (e.g. *Nature immunology*, 2019, 20: 206; *EBioMedicine*, 2019, 41: 73). In this assay, each membrane contains 111 different cytokine antibodies, and each antibody is set in duplicate by the manufacturer. It is commonly practiced that – as both reported in the above mentioned two recent papers and performed in our study – two independent experiments are performed.
- 2) We performed two independent experiments, and from each experiment we could get two quantitative values for each cytokine. The outcomes from the first experiment and the repeated one are shown in the below **Fig. R1** and **Fig. R2**, respectively. We analyzed the outcomes from these two repeats and found that the trend was consistent: i.e. HSD increased the level of key pro-inflammatory cytokines (ICAM-1, IFN- γ and TNF- α) and decreased that of GM-CSF, IL-6 and IL-10 in the serum.
- 3) Despite the consistent outcomes, we choose not to integrate the four readings from the two experiments together, because the outcomes are expressed in relative intensity units. As such, we show the data of one independent experiment in the article.
- 4) The reviewer is right, and we were wrong in our last revision, that this group of data cannot be tested by *t*-test. We have corrected it in the revised paper and present Fig.

R1 as Fig. 3a and Supplementary Fig. 5 in the current submission.

Figure-for-response-only 1 (Fig. R1). Tumour homogenates and serum samples were harvested and mixed at equal quality or equal volumes. Inflammatory cytokines in tumour tissue lysates and serum from tumour-bearing NSD- or HSD-fed mice were assessed by the Proteome Profiler Mouse XL Cytokine Array. (a) The images of inflammatory cytokines detected in tumour tissue lysates and serum. (b) The signal intensity of arrays was analysed using densitometry, and relative intensity (NSD versus HSD) of individual cytokines was analysed after normalizing to the positive controls on the same membrane. Each dot represents the technical replication of one phenotype.

Figure-for-response-only 2 (Fig. R2). See legends for Fig. R1 for the methodology.

Q2. Another example was the answer to Q1. How can it happen that a value of 45.7 is suddenly 39.8? How can it be that the previous quantification was not representing this value? Again, the excuse "due to our negligence" is not a good scientific procedure. Moreover, the new quantification does not visualize a value of 39.8? You mention that you repeated experiments. It seems that you still have the same n-number. The material used is not really suitable for repeating experiments. In the best case, a re-analysis is doable. It is not the duty of a reviewer to be a detective to find out what was done. You NEED to report details that one can understand it.

A2: We understand the reviewer's concern and wish to clarify the changes below:

- 1) For the convenience of description, we refer to each version of our submission as:
 - V1: our original submission, Jun 2018
 - V2: the first revision, Jan 2019
 - V3: the second revision, May 2019
 - V4: the current revision in submission, Aug 2019
- 2) In V2, we for the first time provided the data of ki67 staining, as one experiment suggested by the reviewer. We have now checked the original data used to generate this set of results and found that the value should be 39.8. The number 45.7 was incorrectly presented. However, in our submission V3, neither 39.8 nor 45.7 was used. This was because, as the reviewer wisely pointed out, the methodology used for FACS in preparing V2 was problematic – we failed to gate out the dead cells in processing the data in V2. So, we repeated the whole experiment in preparing V3 of this manuscript.
- 3) In preparing V3, we adopted the correct methodology and gated out the dead cells (FVD eFluor 520, eBioscience). In doing this experiment, we also had 10 mice in each group to ensure the consistency of the experiment and validity of the data. The data, as shown in **Fig. R3** below, suggest that the proliferation of CD8⁺ T cells within

tumour is greater in HSD-treated animals than in NSD-treated animals – the trend is consistent with the old (and inaccurate) data.

Figure-for-response-only 3 (Fig. R3). The flow cytometry analysis of Ki67 expression after gating on CD8+ T cells in the 4T1 tumour tissues. (a, b) The flow cytometry analysis of NSD and HSD group in one independent experiment; (c) The expression of Ki67 in CD8+ T cells. Two-tailed Student's t-test; n = 10 mice per group; *p<0.05 vs. the NSD group.

4) Using the same and correct methodology, we repeated this experiment (another biological repeat (**Fig. R4**). The results in **Fig. R3** and **Fig. R4** are consistent, but are not suitable to be integrated for presentation due to the nature of flow cytometry. Therefore, the data in **Fig. R3** were presented in V3 (Fig. 5e, f) and the current V4 submission (Fig. 5e, f).

Figure-for-response-only 4 (Fig. R4). See legends for Fig. R3.

- 5) As explained above, we always set 10 mice in each group for the two groups (NSD and HSD) in each independent experiment. We had two independent experiments in total. Thus, we indicate n=10 for each representative experiment.

Q3. In fig. 3d-e, the dotted values differ between the last and the current submission. Again, there is no explanation given why this is the case. This habit is from my perspective unacceptable.

A3: This issue occurred also due to our failure in gating out the dead cells in the submission V2. Our clarification is below:

- 1) In the V2 submission, the dotted values for PMN-MDSCs and M-MDSCs in the blood, spleen and tumour (Fig. 3d-e, original) are shown in the **Fig R5 a-d** below.

Figure-for-response-only 5 (Fig. R5). After NSD or HSD for 16 days, the representative flow cytometry analysis (a-c) and the frequency of PMN-MDSCs and M-MDSCs (d) in the blood, spleen and tumour tissues from 4T1 tumour-bearing mice were assessed within total CD45⁺ cells. Two-tailed Student's t-test; n = 10 mice per group; *p<0.05 vs. the NSD group.

- 3) However, as the reviewer wisely pointed out, the authors should have gated out the dead cells to analyze the MDSCs subpopulations. Accordingly, we performed new experiments and analyzed the results after gating out the dead cells. The new results, as shown in **Fig. R6a-d** below, demonstrate that HSD reduced the population of

M-MDSCs but did not affect the the percentage of PMN-MDSCs.

Figure-for-response-only 6 (Fig. R6). See legends for Fig. R5.

4) Therefore, to better explain the function of HSD, the old Fig. 3 d-e (Fig. R5 a-d) in

the V2 submission was replaced by new data (**Fig. R6 a-d**) in V3 and the current V4 submissions. The dotted values in V2 differ from those in the V3 and V4.

- 5) Finally, we thank the reviewer for all such massively important help. We once again apologize for all the errors in cytometry methodology and data processing. We can confirm that we are now providing convincing data based on repeated experiments. We hope that, by providing the detailed procedure for data processing, we can explain all the changes that may have caused confusion. We are also providing all the original data in the Data Source files in this submission.

Reviewers' comments:

Reviewer #4 (Remarks to the Author):

The manuscript by Dong et al uncovers some interesting biology regarding the role of a high salt diet on tumor progression. The authors interrogate this using two separate xenograft models in two distinct backgrounds B6 and Balb/c. Interestingly, they propose that a high salt diet through effects on MDSCs reduces tumor progression in part by activating an immune response. Overall, I find the manuscript interesting and the main conclusions sound in the version that I read. However, some addressable concerns, also noted by previous reviewers, dampens my enthusiasm as listed below:

1) The authors use two-tailed student's t-tests to assess differences between groups. This statistical test is not optimal given that it is for large sample sizes and assumes normality of the data, which isn't the case for some of the experiments. A more appropriate test is the two-tailed Wilcoxon rank-sum tests. With this test, some of the experiments are likely underpowered.

2) Figure panels 5e and f are not convincing. Perhaps the authors can sort CD4 and CD8 + T cells and assess Ki67 by IF on cytopun cells or measure EDU/BrDU incorporation.

3) The authors implicate NFAT5 as mediating the effects of HSD on tumor growth. However, the in vivo experiments were tested with a lentiviral Ly-6C-siNFAT5 construct. As far as I can tell very little in vivo validation was performed in immune cell subsets. Is NFAT5 deleted in tumor MSDCs? Is NFAT5 deleted in other immune or epithelial subsets? IHC/IF validation for NFAT5 in tumors would be relatively simple. Similarly, anti-Gr1 antibody depletion may eliminate more than MSDCs, such as early hematopoietic progenitors--this should be discussed.

4) Diet based studies typically list the main constituents of the diet in tables.

5) The authors need to provide a citation for the Balb/c nu-nu model as I could not find it in the manuscript.

Response to Reviewer #4:

Q1. The authors use two-tailed student's t-tests to assess differences between groups. This statistical test is not optimal given that it is for large sample sizes and assumes normality of the data, which isn't the case for some of the experiments. A more appropriate test is the two-tailed Wilcoxon rank-sum tests. With this test, some of the experiments are likely underpowered.

A1. We thank the reviewer for this excellent suggestion. For the data with a large sample size, differences between two groups were re-evaluated using the two-tailed Wilcoxon rank-sum tests – including Figure 1, Figure 2, Figure 3b-g, Figure 4h, Figure 5, Supplementary Figure 1, Supplementary Figure 2, Supplementary Figure 3, Supplementary Figure 4, Supplementary Figure 6, Supplementary Figure 11 and Supplementary Figure 13. All the experiments re-evaluated using the two-tailed Wilcoxon rank-sum tests are statistically significant.

Q2. Figure panels 5e and f are not convincing. Perhaps the authors can sort CD4 and CD8⁺ T cells and assess Ki67 by IF on cytopun cells or measure EDU/BrDU incorporation.

A2: We are pleased to provide the experimental details below:

- 1) We sorted CD4 and CD8⁺ T cells from tumour tissues of the tumour-bearing mice with NSD or HSD treatment, by using Anti-Mouse CD4 Magnetic Particles – DM and Anti-Mouse CD8a Particles – DM. We carried out the experiments following the manufacturer's instructions, with all steps performed at 4 °C, and assessed their proliferation using Ki67 antibody by IF;
- 2) We found that the proportion of Ki67⁺ T cells in total T cells was higher in HSD-treated than in NSD-treated mice, and the expression of Ki67 in most T cells was high in HSD-treated animals, compared to the NSD-treated animals

(Supplementary Figure 11 in the revised paper).

- 3) These results demonstrated that HSD enhanced the T cells' proliferation.

Q3. The authors implicate NFAT5 as mediating the effects of HSD on tumor growth. However, the *in vivo* experiments were tested with a lentiviral Ly-6C-siNFAT5 construct. As far as I can tell very little *in vivo* validation was performed in immune cell subsets. Is NFAT5 deleted in tumor MSDCs? Is NFAT5 deleted in other immune or epithelial subsets? IHC/IF validation for NFAT5 in tumors would be relatively simple. Similarly, anti-Gr1 antibody depletion may eliminate more than MSDCs, such as early hematopoietic progenitors--this should be discussed.

A3. These are excellent questions and our responses are below.

- 1) Yes, we validated the effects of NFAT5 silencing in immune cell subsets *in vivo*: a lentivirus carrying NFAT5 siRNA under the Ly-6C promoter (NFAT5 siRNA-LV) or control siRNA-LV was intravenously injected into tumour-bearing mice every two days for 16 days. Then, M-MDSCs, PMN-MDSCs, macrophages and CD3⁺ T cells were isolated from tumour tissues, and the expression of NFAT5 was examined by RT-qPCR and Western blotting.
- 2) The results showed that the expression of NFAT5 was down-regulated in M-MDSCs and macrophages, but it was not changed in PMN-MDSCs or CD3⁺ T cells (Supplementary Figure 16).
- 3) The epithelial cells do not express Ly-6C. So, a lentiviral Ly-6C-siNFAT5 construct did not affect the expression of NFAT5 in epithelial cells.
- 4) In accordance, the Supplementary Figure 15a in the last revision became **Supplementary Figure 16** in the new revision.
- 5) We agree with the reviewer that anti-Gr1 antibody depletion may eliminate more cells than MSDCs; however, it is widely agreed that this antibody is an effective tool for eliminating MDSCs in the field of cancer [Immunity, 2017,

Nature communications, 2018]. In fact, anti-Gr1 antibody depletion could eliminate Gr1⁺ cells, which may include immature myeloid cells (IMCs), MDSCs and neutrophils. IMCs with the same phenotype as the MDSCs are continually generated in the bone marrow of healthy individuals and differentiate into mature myeloid cells, whereas this process can be redirected toward the differentiation of pathological MDSCs in cancer [Nature Review Immunology, 2009; Nature Review Immunology, 2012]. MDSCs are a heterogeneous population of IMCs and myeloid progenitors that negatively regulate immune responses [PNAS, 2014]. In addition, PMN-MDSC are pathologically activated neutrophils that are critically important for the regulation of immune responses in cancer [Nature, 2019]. Therefore, anti-Gr1 antibody was widely used for eliminating MDSCs in the field of cancer [Immunity, 2017, Nature communications, 2018].

- 6) However, early hematopoietic progenitors might not be a big concern in this study. Usually, the hematopoietic progenitors could differentiate into lineage-committed progenitors, e.g., the common myeloid progenitors and granulocyte-monocyte progenitors. These progenitors in the bone marrow and peripheral blood can differentiate into IMCs, and IMCs migrate to different peripheral organs, where they differentiate into macrophages, dendritic cells, or granulocytes [PNAS, 2014; Annual Review of Immunology, 2003]. In cancer patients and tumour-bearing experimental animals, hematopoietic stem and progenitor cells could form a metastasis-conducive microenvironment, contributed to experimental metastasis, and differentiate into MDSCs [Cancer research, 2017]. In addition, the hematopoietic progenitors do not express Gr1 markers [Annual Review of Immunology, 2003], and anti-Gr1 antibody depletion may not affect the function of hematopoietic progenitors.
- 7) Therefore, the elimination of Gr1⁺ cells by anti-Gr1 antibody is mainly MDSCs, although anti-Gr-1 antibody may also reduce the number of IMCs and neutrophils, which has been discussed in the section of discussion in the

revised paper. We highly appreciate the reviewer for this constructive suggestion.

Q4. Diet based studies typically list the main constituents of the diet in tables.

A4: Thank you for this reminder. We have now provided the ingredients and caloric content in the NSD and HSD in the revised **Supplementary Table 3**.

Q5. The authors need to provide a citation for the Balb/c nu-nu model as I could not find it in the manuscript.

A5: We thank the reviewer for this timely reminder. We have added the literature for the Balb/c nu-nu model as **Ref. 33** and **34** in the revised paper.

REVIEWERS' COMMENTS:

Reviewer #4 (Remarks to the Author):

The authors have adequately addressed my concerns.